# Automatic Reward Shaping from Confounded Offline Data

**Mingxuan Li** [1]   **Junzhe Zhang** [2]   **Elias Bareinboim** [1]

## Abstract

Reward shaping has been demonstrated to be an effective technique for accelerating the learning process of reinforcement learning (RL) agents. While successful in empirical applications, the design of a good shaping function is less well understood in principle and thus often relies on domain expertise and manual design. To overcome this limitation, we propose a novel automated approach for designing reward functions from offline data, possibly contaminated with the unobserved confounding bias. We propose to use causal state value upper bounds calculated from offline datasets as a conservative optimistic estimation of the optimal state value, which is then used as state potentials in Potential-Based Reward Shaping (PBRS). When applying our shaping function to a model-free learner based on UCB principles, we show that it enjoys a better gap-dependent regret bound than the learner without shaping. To the best of our knowledge, this is the first gap-dependent regret bound for PBRS in model-free learning with online exploration. Simulations support the theoretical findings.

## 1. Introduction

Reward shaping is an effective technique for improving sample efficiency in reinforcement learning. It augments the original environment reward with extra learning signals such that the learner is guided towards high-rewarding states in the environment. Though if not designed properly, there is a risk of misleading the agent towards suboptimal policies (Saksida et al., 1997; Randløv & Alstrøm, 1998). **P**otential **B**ased **R**eward **S**haping (PBRS, Ng et al. (1999)) solves this problem by constructing the shaping functions as the difference between state potentials. In this way, the

set of optimal policies after reward shaping is guaranteed to be unchanged compared with the one without shaping. It has been followed in many practical applications since then.

The core of PBRS is the design of potential functions, which denote the potential of a state or how good a state is. First, people generally rely on domain expertise to compose such potential functions. This dependence on experts' knowledge could make designing potential functions expensive and time-consuming. One may even argue these extensive human efforts are against the promise of AI that domain experts should be free from handcrafting solutions. Moreover, this designing process endures risks of misspecification due to human biases, which in turn could slow down the training substantially or lead to erroneous agents (Ng et al., 1999; Randløv & Alstrøm, 1998; Pan et al., 2022). An ongoing line of research exists to simultaneously automate the process of learning the shaping function and training the online agent (Pathak et al., 2017; Yuan et al., 2023; Raileanu & Rocktäschel, 2020; Devidze et al., 2022; Zou et al., 2019; Ma et al., 2024). However, these methods involve nontrivial optimization procedures relying on prior parametric knowledge over the potential function. The risks of misspecification persist. In other words, designing potential functions is still a significant obstacle when applying PBRS.

An alternative strategy is to learn the shaping function from previous offline data, possibly collected by different behavior policies or observing human operators interacting with the environment (Brys et al., 2015; Mezghani et al., 2022; Zhang et al., 2024). In their seminal work, Ng et al. (1999) noted that the optimal state value is a good candidate for potential functions, characterizing the "state potentials" in the underlying environment. In the field of reinforcement learning, the problem of evaluating optimal state value functions from past offline data has been studied under the rubrics of off-policy learning or batch learning (Sutton, 2018; Levine et al., 2020). Several algorithms and methods have been proposed, including Q-learning (Watkins, 1989; Watkins & Dayan, 1992), importance sampling (Swaminathan & Joachims, 2015; Jiang & Li, 2016), and temporal difference (Precup et al., 2000; Munos et al., 2016). These algorithms rely on the critical assumption that there is no unobserved confounding (NUC) in the offline data (Murphy, 2003; 2005). One might be able to enforce this assumption by deliberately controlling the behavior policies generat-

---

[1]Causal AI Lab, Columbia University, New York, USA [2]Department of Electrical Engineering and Computer Science, Syracuse University, New York, USA. Correspondence to: Mingxuan Li <ml@cs.columbia.edu>.

*Proceedings of the $42^{nd}$ International Conference on Machine Learning*, Vancouver, Canada. PMLR 267, 2025. Copyright 2025 by the author(s).

ing the data. However, in many practical applications, the NUC condition can be difficult to enforce and, consequently, does not necessarily hold. In these cases, directly applying standard off-policy learning could introduce bias in the estimation, leading to misspecified shaping functions. For example, when some input states to the behavior policy are not fully observed, these states could become unobserved confounders, introducing spurious correlations in the offline data. When NUC is violated, the effects of candidate policies are generally not *identifiable*, i.e., the model assumptions are insufficient to uniquely determine the value function from the offline data, regardless of the sample size (Pearl, 2009; Zhang & Bareinboim, 2019).

This paper addresses the challenges of confounding bias in designing potential functions for reward shaping. We study the problem of constructing reward-shaping functions automatically via confounded offline data from a causal perspective. We focus on the environment of Confounded Markov Decision Processes (CMDPs) where unobserved confounders generally exist at each stage of decision-making Zhang & Bareinboim (2022); Tennenholtz et al. (2020). Under CMDPs, we demonstrate the difficulties of manually designing proper reward shaping functions with the presence of unobserved confounding, due to the mismatch in the observed input states. Then we show how one can extrapolate informative shaping functions from confounded offline data using partial causal identification techniques (Manski, 1989). Finally, we develop novel online reinforcement learning algorithms that can leverage the derived shaping functions to improve the learning performance in terms of the sample complexity. More specifically, our contributions are summarized as follows:

- We propose the first theoretically justified (Thm. 3.1), data-driven method (Algo. 2) for learning reward shaping functions from confounded offline data.
- We introduce a model-free UCB algorithm that can improve performance by leveraging confounded offline data using the derived shaping functions (Algo. 1).
- We derive a novel gap-dependent regret bound for the proposed UCB algorithm. Our analysis reveals how and under what condition (Def. 4.1) the derived shaping functions affect the learning efficiency of future online learners (Thm. 4.5).

Due to limited space, related work, experiment details and all the proof are provided in Apps. A, D and H, respectively.

**Notations.** We will consistently use capital letters ($V$) to denote random variables, lowercase letters ($v$) for their values, and cursive $\mathcal{V}$ to denote their domains. Fix indices $i, j \in \mathbb{N}$. We use bold capital letters ($\boldsymbol{V}$) to denote a set of random variables and let $|\boldsymbol{V}|$ denote its cardinality of set $\boldsymbol{V}$. Finally, $\mathbf{1}_{\boldsymbol{Z}=\boldsymbol{z}}$ is an indicator function that returns 1 if event $\boldsymbol{Z} = \boldsymbol{z}$ holds true; otherwise, it returns 0.

## 2. Challenges of Designing Reward Shaping in the Face of Unobserved Confounders

We will focus on a sequential decision-making setting in the Markov Decision Process (MDP, Puterman (1994)) where the agent intervenes on a sequence of actions $X_1, \ldots, X_H$ in order to optimize the cumulative return over reward signals $Y_1, \ldots, Y_H$; $H \in \mathbb{N}$ is a finite horizon.

Standard MDP formalism focuses on the perspective of the learners who could actively intervene in the environment. Consequently, the data collected from randomized experiments is free from the contamination of unobserved confounding bias and is generally assumed away in the model. However, when considering offline data collected by passive observation, the learner may not necessarily have deliberate control over the behavioral policy generating the data. Consequently, this could lead to confounding bias in various decision-making tasks, including off-policy learning (Kallus & Zhou, 2018; Lu et al., 2023; Zhang & Bareinboim, 2024), and imitation learning (Zhang et al., 2020; Kumor et al., 2021; Ruan et al., 2024). In this paper, we will consider an extended family of MDPs explicitly modeling the presence of unobserved confounders when generating offline data.

**Definition 2.1.** A Confounded Markov Decision Process (CMDP) $\mathcal{M}$ is a tuple of $\langle \mathcal{S}, \mathcal{X}, \mathcal{Y}, \mathcal{U}, H, \mathbb{F}, \mathbb{P} \rangle$ where,

- $\mathcal{S}, \mathcal{X}, \mathcal{Y}$ are, respectively, the space of observed states, actions, and rewards;
- $\mathcal{U}$ is the space of unobserved exogenous noise;
- $H \in \mathbb{N}$ is a finite horizon;
- $\mathbb{F}$ is a set consisting of the transition function $\tau_h : \mathcal{S} \times \mathcal{X} \times \mathcal{U} \mapsto \mathcal{S}$, behavioral policy $\beta_h : \mathcal{S} \times \mathcal{U} \mapsto \mathcal{X}$, and reward function $r_h : \mathcal{S} \times \mathcal{X} \times \mathcal{U} \mapsto \mathcal{Y}$ for every time step $h = 1, \ldots, H$;
- $\mathbb{P}$ is a set of distributions $P_h$ over the unobserved domain $\mathcal{U}$ for every time step $h = 1, \ldots, H$.

Consider a demonstrator agent interacting with a CMDP. For every time step $h = 1, \ldots, H$, the nature draws an exogenous noise $U_h$ from the distribution $P(\mathcal{U})$; the demonstrator performs an action $X_h \leftarrow \beta_h(S_h, U_h)$, receives a subsequent reward $Y_h \leftarrow r_h(S_h, X_h, U_h)$, and moves to the next state $S_{h+1} \leftarrow \tau_h(S_h, X_h, U_h)$. The observed trajectories of the demonstrator (from the learner's perspective) are thus summarized as the observational distribution $P(\bar{\boldsymbol{X}}, \bar{\boldsymbol{S}}, \bar{\boldsymbol{Y}})$.[1]

In the data-generating process described above, for every time step $t$, the exogenous noise $U_h$ becomes an unobserved confounder affecting the action $X_h$, reward $Y_h$, and next state $S_{h+1}$ simultaneously. Therefore, CMDP is also referred to MDP with Unobserved Confounders (MDPUC, Zhang & Bareinboim (2022)) and is a subclass of Con-

---

[1] We will consistently use $\bar{\boldsymbol{X}}, \bar{\boldsymbol{S}}, \bar{\boldsymbol{Y}}$ to represent sequences $\{X_1, \ldots, X_H\}, \{S_1, \ldots, S_H\}$ and $\{Y_1, \ldots, Y_H\}$

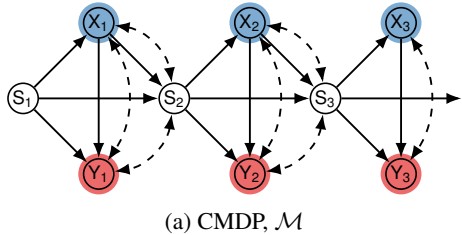

(a) CMDP, $\mathcal{M}$

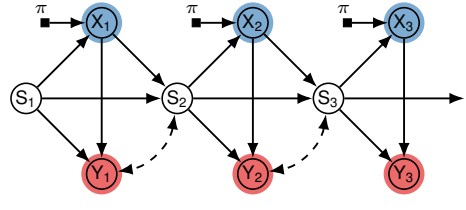

(b) CMDP - Online Learning, $\mathcal{M}_\pi$

*Figure 1.* (a) Causal diagram of the CMDP modeling the shaping function designing process; (b) Causal diagram of the CMDP modeling the online learning process under policy $\text{do}(\pi)$.

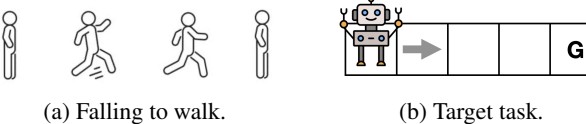

(a) Falling to walk.      (b) Target task.

*Figure 2.* Walking Robot Example.

founded Partially Observed MDP (Shi et al., 2022; Miao et al., 2022; Bennett & Kallus, 2024) where Markov property holds. The following example, inspired by human walking dynamics (O'Connor & Kuo, 2009; Wang & Srinivasan, 2014), demonstrates an instance of CMDP.

**Example 1** (Walking Robot). Consider a scenario where a robot learns to walk across a hallway. The agent can take two actions, small-step and big-step, denoted as $X_h = 0$ and $X_h = 1$, respectively. Whether the agent can move forward ($L_{h+1}$) depends on both the step size, $X_h$, and the current body stability status, $F_h$. It's defined as $L_{h+1} \leftarrow L_h + 1[(\neg F_h \wedge (X_h = U_h)) \vee (F_h \wedge \neg X_h)]$. The body stability is modeled as $F_{h+1} \leftarrow \neg X_h$ if the body is in stable status $F_h = 1$ or $F_{h+1} \leftarrow \neg X_h \oplus U_h$ if it is not stable $F_h = 0$ and needs to adjust step size accordingly. The required step size to stay stable is modeled as a uniform random noise $U_h$. The agent receives a reward $Y_h = 1$ if it reaches the goal location or it moves forward. It receives a reward $Y_h = -1$ if it cannot stabilize the body accordingly. Formally, $Y_h \leftarrow 1$ if $L_{h+1} = G$ otherwise $Y_h \leftarrow 1[(\neg F_h \wedge X_h = U_h) \vee (F_h \wedge \neg X_h)] - 1[\neg F_h \wedge X_h \oplus U_h]$.

In this example, body stability $F_h$ affects whether the robot can step forward. We will consider offline data generated by two demonstrators. When the robot is currently unstable $F_h = 0$, a competent demonstrator following a behavioral policy $\beta_h^{(1)} : X_h \leftarrow U_h$ takes a step in the size of exactly the latent noise $U_h$. As a result, it will always transit from being unstable $F_h = 0$ to a stable status $F_{h+1} = \neg X_h \oplus U_h = \neg U_h \oplus U_h = 1$. On the contrary, an incompetent demonstrator following $\beta_X^{(2)} : X_h \leftarrow \neg U_h$ will always attempt to remain unstable $F_{h+1} = \neg X_h \oplus U_h = U_h \oplus U_h = 0$ even when the opposite is preferable. ∎

Fig. 1a shows the graphical representation (i.e., causal diagram) describing the generative process generating the

offline data in CMDPs. More specifically, nodes represent observed variables $X_h, S_h, Y_h$, and arrows represent the functional relationships $\beta_h, \tau_h, r_h$ among them. The exogenous noise $U_h$ is often not explicitly shown. However, bi-directed arrows $X_h \leftrightarrow Y_h$ and $X_h \leftrightarrow S_{h+1}$ indicate the presence of an unobserved confounder (UC) $U_h$ affecting $X_h, Y_h$ and $S_{h+1}$, simultaneously. These bi-directed arrows characterize the spurious correlations among action $X_h$, reward $Y_h$, and state $S_{h+1}$ in the offline data, violating the NUC condition. Such violations could lead to challenges in evaluating value function and reward shaping, which we will discuss in the next section.

### 2.1. Potential-Based Reward Shaping

A policy $\pi$ in a CMDP $\mathcal{M}$ is a sequence of decision rules $\pi_h : \mathcal{S} \mapsto \mathcal{X}$, for every step $h = 1, \ldots, H$, mapping from state to action. Similarly, $\pi_h(x_h \mid s_h)$ is a stochastic policy mapping from state space $\mathcal{S}$ to a distribution over action space $\mathcal{X}$. An intervention $\text{do}(\pi)$ is an operation that replaces the behavioral policy $\beta_h$ in model $\mathcal{M}$ with the decision rule $\pi_h$ for every step $h$. Let $\mathcal{M}_\pi$ be the submodel induced by intervention $\text{do}(\pi)$. Fig. 1b shows the graphical representation of the data generating process in the submodel $\mathcal{M}_\pi$; bi-directed arrows are now removed.

The interventional distribution $P_\pi(\bar{\boldsymbol{X}}, \bar{\boldsymbol{S}}, \bar{\boldsymbol{Y}})$ is defined as the joint distribution over observed variables in $\mathcal{M}_\pi$, i.e.,

$$P_\pi(\bar{\boldsymbol{x}}, \bar{\boldsymbol{s}}, \bar{\boldsymbol{y}}) = P(s_1) \prod_{h=1}^{H} \Bigg( \pi_h(x_h \mid s_h) \\ \mathcal{T}_h(s_h, x_h, s_{h+1}) \mathcal{R}_h(s_h, x_h, y_h) \Bigg) \quad (1)$$

where the transition distribution $\mathcal{T}_h$ and the reward distribution $\mathcal{R}_h$ are given by, for $h = 1, \ldots, H$,

$$\mathcal{T}_h(s, x, s') = \int_u \mathbf{1}_{s' = \tau_h(s, x, u)} P_h(u) \quad (2)$$

$$\mathcal{R}_h(s, x, y) = \int_u \mathbf{1}_{y = r_h(s, x, u)} P_h(u) \quad (3)$$

For convenience, we write the reward function $\mathcal{R}_h(s, x)$ as the expected value $\sum_y y \mathcal{R}_h(s, x, y)$.

For reinforcement learning tasks, the agent's goal is to learn an optimal policy $\pi^*$ maximizing the cumulative reward in CMDP $\mathcal{M}$, i.e., $\pi^* = \arg\max_\pi E_\pi \left[ \sum_{h=1}^H Y_h \right]$. In analysis, we also evaluate the state value function $V_h^\pi(s) = E_\pi \left[ \sum_{t=h} Y_t \mid S_h = s \right]$ following a policy $\pi$ for every step $h = 1, \ldots, H$. A state-action value function is defined as $Q_h^\pi(s, x) = E_\pi \left[ \sum_{t=h} Y_t \mid S_h = s, X_h = x \right]$.

Reward shaping is a popular line of techniques for incorporating domain knowledge during policy learning. Common approaches such as Potential-Based Reward Shaping (PBRS, Ng et al. (1999)) add supplemental signals to the reward function so that it would be easier to learn in future downstream tasks without affecting the optimality of the learned policy. The following proposition establishes the validity of PBRS in CMDP with a finite horizon.

**Proposition 2.2.** *For a CMDP $\mathcal{M}$, let $\mathcal{M}'$ be a CMDP obtained from $\mathcal{M}$ by replacing the reward function with the following, for every time step $h = 1, \ldots, H$,*

$$r_h' := r_h + \phi_h(S_{h+1}) - \phi_h(S_h), \tag{4}$$

*where $r_h$ is the reward function in $\mathcal{M}$; $\phi_h(\cdot) : \mathcal{S} \mapsto \mathbb{R}$ is a real valued potential function and $\phi_H(s) = 0$. Then,*

.

*That is, every optimal policy in $\mathcal{M}$ will also be an optimal policy in $\mathcal{M}'$, and vice versa.*

In other words, PBRS modifies the reward function in the system to encourage the learning agent to visit states with high potential in future return. At the same time, the agent is penalized for visiting states with low potential. More importantly, optimal policies remain invariant across this shaping process, i.e., every optimal policy in the after-shaping model $\mathcal{M}'$ is guaranteed to be optimal in the original model $\mathcal{M}$.

Designing the potential function is a central problem when applying PBRS. The optimal state value functions $V_h^*(s) = V_h^{\pi^*}(s)$ (Ng et al., 1999) are a good candidate for measuring state potential. When the NUC condition holds, it is possible to compute such optimal value functions using standard off-policy methods (Sutton, 2018). However, when unobserved confounding generally exists, evaluating value functions from confounded data could introduce bias in the shaping process, promoting states with low potential. The following example demonstrates such challenges.

**Example 2** (Confounded Potential Functions)**.** Consider again the Walking Robot in Example 1. Computing the optimal state value function $V_h^*(L_h, F_h)$ from the ground-truth model $\mathcal{M}$ gives, for every step $h = 1, \ldots, H$,

$$V_h^*(L_h = 0, F_h = 0) = 5.0, \; V_h^*(L_h = 0, F_h = 1) = 5.5$$

If one sets the potential function $\phi_h := V_h^*$, the reward shaping will add the supplemental signal in favor of transitioning from an unstable state $F_h = 0$ to a stable state $F_{h+1} = 1$,

$$\phi_h(L_{h+1} = 0, F_{h+1} = 1) - \phi_h(L_h = 0, F_h = 0) = 0.5$$

On the other hand, if one applies standard off-policy methods using offline data generated by the competent behavioral policy $\beta_h^{(1)}$, it leads to the following value function,

$$V_h^{(1)}(L_h = 0, F_h = 0) = 10, \; V_h^{(1)}(L_h = 0, F_h = 1) = 10$$

This seems to suggest that being stable does not improve the robot's performance. If we set the potential function $\phi_h := V_h^{(1)}$, the reward shaping will not encourage the robot to be stable. More interesting, if we compute the value function from offline data generated by the incompetent behavioral policy $\beta_h^{(2)}$, we have the following

$$V_h^{(2)}(L_h = 0, F_h = 0) = -23, \; V_h^{(2)}(L_h = 0, F_h = 1) = -42$$

Being stable $F_h = 1$ appears to lead to lower future returns. If one sets the potential function $\phi_h := V_h^{(2)}$, the reward shaping will even penalize the robot to transit from being unstable $F_h = 0$ to being stable $F_{h+1} = 1$, which contradicts the underlying system dynamics. ∎

The above example shows that when unobserved confounders generally exist, naively computing potential functions from offline data could lead to biased evaluation. Shaping reward with a biased potential function could lead to sub-optimal performance, encouraging undesirable behaviors. Indeed, when the NUC does not hold, the transition function $\mathcal{T}$ and reward function $\mathcal{R}$ are generally not uniquely discernible from the offline data, regardless of the sample size (Pearl, 2009). For the remainder of this paper, we will introduce a robust procedure to design potential functions from confounded offline data, and how to leverage these potential functions for future online learning tasks.

## 3. Robust Potential Functions from Confounded Offline Data

Our goal is to upper bound the optimal state value for online interventional agents, the ones that cannot utilize confounders, from potentially confounded offline datasets. Instead of attempting to identify the underlying reward and transition distribution under confounded datasets, we bound them for every $s, x, s', h \in \mathcal{S} \times \mathcal{X} \times \mathcal{S} \times [H]$ following similar partial identification strategies in (Manski, 1989),

$$\mathcal{T}_h(s, x, s') \leq \widetilde{T}_h(s, x, s') P_h(x|s) + P_h(\neg x|s) \tag{5}$$

$$\mathcal{R}_h(s, x) \leq \widetilde{R}_h(s, x) P_h(x|s) + b P_h(\neg x|s) \tag{6}$$

where $\widetilde{\mathcal{R}}_h(s, x) = \mathbb{E}[Y_h | S_h = s, X_h = x]$, $\widetilde{\mathcal{T}}_h(s, x, s') = P_h(S_{h+1} = s' | S_h = s, X_h = x)$ and $P_h(x|s) = P_h(X_h = $

$x|S_h = s)$ are all empirical estimations from the offline dataset. $b$ is a known upper bound on the reward signal, $Y_h \leq b$. Similar to the bounding approach developed in (Zhang & Bareinboim, 2024), we apply the above bounds to the Bellman Optimal Equation (Sutton & Barto, 2018) and arrive at the following Causal Bellman Optimal Equation that calculates the optimal state value optimistically from the confounded offline datasets.

**Theorem 3.1** (Causal Bellman Optimal Equation). *For a CMDP environment $\mathcal{M}$ with reward $Y_h \leq b, b \in \mathbb{R}$, the optimal value of interventional policies, $V_h^*(s), \forall s \in \mathcal{S}$, is upper bounded by $V_h^*(s) \leq \overline{V}_h(s)$ satisfying the Causal Bellman Optimality Equation, for every step $h = 1, \ldots, H$,*

$$\overline{V}_h(s) = \max_x \Big[ P_h(x|s) \Big( \widetilde{\mathcal{R}}_h(s, x) + \mathbb{E}_{\widetilde{\mathcal{T}}_h}[\overline{V}_{h+1}(s')] \Big)$$
$$+ P_h(\neg x|s) \Big( b + \max_{s'} \overline{V}_{h+1}(s') \Big) \Big] \quad (7)$$

*and $\overline{V}_{H+1}(s) = 0$ for all state $s \in \mathcal{S}$.*

Compared with the original Bellman Optimal Equation, ours accounts for the uncertainty brought by confounders in the offline dataset via an extra term, $b + \max_{s'} \overline{V}_{h+1}(s')$. This represents the best rewards that the agent could have achieved from those "unselected" actions, i.e., $P_h(\neg x|s)$. With the Causal Bellman Optimal Equation, we can robustly upper bound the optimal state values from a confounded offline dataset generated by CMDP $\mathcal{M}$.

When multiple offline datasets from different behavioral policies are available, each of them provides a unique perspective on the true underlying CMDP without necessarily overlapping state space coverage. Since the upper bounds calculated for each state from different datasets are all valid according to Thm. 3.1, taking the minimum yields the tightest estimation for the optimal state value.

**Corollary 3.2** (Unified Causal Optimal Upper Bound). *Let the causal optimal value upper bound estimated from offline datasets $\mathcal{D}^{(i)}$ be $\overline{V}^{(i)}, i = 1, 2, ..., N$, the unified causal optimal upper bound is defined as $\overline{V}_h(s) = \min_{s \in \mathcal{D}_h^{(i)}} \overline{V}_h^{(i)}(s)$ where $s \in \mathcal{D}_h^{(i)}$ means state $s$ is visited at step $h$ in $\mathcal{D}^{(i)}$ and satisfies $V_h^*(s) \leq \overline{V}_h(s)$, for all $s$.*

In actual computation, to avoid over-estimation brought by bootstrapping on the highest value state, we clip the optimistic bonus for the optimal transitions and calculate the value update as follows,

$$\overline{Q}_h^{(i)}(s, x) = P_h^{(i)}(\neg x|s) \Big( b + \min \Big\{ \max_{s^*} V_{h+1}^{(i)}(s^*), \omega_h \Big\} \Big)$$
$$+ P_h^{(i)}(x|s) \Big( \widetilde{R}_h^{(i)}(s, x) + \sum_{s'} \widetilde{T}_h^{(i)}(s, x, s') V_{h+1}^{(i)}(s') \Big) \quad (8)$$

where $\omega_h = (H - h)b$ is the maximum possible next state value an optimistic transition should receive in an episodic

CMDP with horizon $H$. For state-action pairs that are not visited in the dataset, we will skip those state-action entries directly.[2] To summarize, the overall algorithmic procedure to calculate causal bounds runs as standard value iteration (Sutton, 2018) except that we update the state values backwards in time, i.e., from $\overline{V}_{h+1}$ to $\overline{V}_h$ by Eq. (8), skip unvisited state-action pairs and take the minimum according to Corol. 3.2 at last. Algo. 2 in App. C shows the full pseudo-code for approximating the optimal value upper bound from offline datasets. Now we can calculate the proposed state potentials for the Walking Robot problem and verify if it upper bounds the true optimal state values.

**Example 3** (Potential Functions Calculated for Robot Walking). Recall that running value iteration in the interventional policy space, we have the optimal state value for Example 1,

$$V_h^*(L_h = 0, F_h = 0) = 5.0, \ V_h^*(L_h = 0, F_h = 1) = 5.5.$$

As we have shown, directly calculating state value from offline datasets cannot yield informative potential functions. While with Corol. 3.2, we can extract the following potential functions from the same offline datasets,

$$\phi_h(L_h = 0, F_h = 0) = 9.0, \ \phi_h(L_h = 0, F_h = 1) = 9.5,$$

which perfectly upper bound the optimal state values and encourages the agent to stay in stable status. From the interventional policy space optimal value and our potential functions, we see that for interventional agents, stability is always preferred, which can be achieved via taking small steps ($X_t = 0$) every time. This also aligns well with the intuition that people glide with small steps when walking on slippery surfaces. See App. B for the the full confounded state values, optimal values and calculated bounds. ∎

## 4. Efficient Online Learning with PBRS

In this section, we will apply the derived potential function from the confounded offline data to improve the agent's performance in an online learning task. For a CMDP $\mathcal{M}$, let $\phi_h$, $h = 1, \ldots, H$, be the potential functions derived in the previous section. We first apply reward shaping and obtain an augmented CMDP $\mathcal{M}'$ following the procedure described in Prop. 2.2.[3]

In the augmented CMDP $\mathcal{M}'$, an online learning agent attempts to learn an optimal policy $\pi^*$ by performing interventions for repeated episodes $k = 1, \ldots, K$. For every episode $k$, the agent picks a policy $\pi^k$, performs intervention $\text{do}(\pi^k)$, and receives subsequent outcomes. The cumulative regret after $K > 0$ episodes of interventions in the augment

---

[2]For a state not visited in any of the offline datasets, we can simply set the bound optimistically as $\overline{V}_h(s) = (H - h)b$.

[3]This can be done by adding supplemental signal $\phi_h(S_{h+1}) - \phi_h(S_h)$ to every observed reward $Y_t$.

CMDP $\mathcal{M}'$ is defined as the sum of the gap between the optimal value function $V_1^*$ following an optimal $\pi^*$ and the value function $V_1^{\pi^k}$ induced by policy $\pi^k$. Formally,

$$\mathbb{E}_{\mathcal{M}'}\left[\text{Regret}(K)\right] = \mathbb{E}_{\mathcal{M}'}\left[\sum_{k=1}^{K} V_1^*(S_1^k) - V_1^{\pi_k}(S_1^k)\right]$$

A desirable property for the agent is to achieve a sublinear regret $\mathbb{E}_{\mathcal{M}'}\left[\text{Regret}(K)\right] = o(K)$.[4] This means that it is able to converge to an optimal policy $\pi^*$.

We propose an online learning algorithm based on a model-free learner, Q-UCB (Jin et al., 2018), to leverage the potential function $\phi$ extrapolated from offline data. Details of the algorithm is described in Algo. 1. Compared with the original Q-UCB, we make a few modifications for Q-UCB to work with PBRS: (1) Zero initializing the Q-values; (2) Using potential function dependent UCB bonus and value clipping; and finally, (3) Incorporating shaped reward during learning updates. See also App. F for the pseudo-code of the vanilla Q-UCB. As in the original Q-UCB, we have visitation counter, $N_h(s_h, x_h)$, for the state-action pair $(s_h, x_h)$ at step $h$. Note that the UCB bonus uses $t$ as its denominator under the square root, which is the visitation count of $(s_h, x_h)$ at the beginning of episode $k$. The counter is updated after assigning $t$, $N_h(s_h, x_h) \leftarrow N_h(s_h, x_h) + 1$. We use the same adaptive learning rate as $\alpha_t = \frac{H+1}{H+t}$ and define $\iota = \log(|\mathcal{S}||\mathcal{X}|T/p)$ where $p' = 2p$ is the probability of the event when the difference between learned and the optimal Q-values is bounded.

As we shall see later in experiments (Fig. 4), using confounded values in Q-UCB Shaping usually impairs the training efficiency rather than helping. While using potential functions that upper bound the optimal state values, learning efficiency is significantly boosted. This observation also resonates with the optimism in the face of uncertainty (OFU) heuristics in reinforcement learning (Gupta et al., 2022; Brafman & Tennenholtz, 2002; Kearns & Singh, 2002). We formalize this observation as the following condition.

**Definition 4.1** (Conservative Optimism). The learned potential function is conservatively optimistic if it satisfies, $V_h^*(s) \leq \phi_h(s) \leq H$ for all $S, h \in \mathcal{S} \times [H]$ where

$$V_h^*(s) = \max_{\pi \in \Pi} \mathbb{E}_{\mathcal{M}_\pi}\left[\sum_{h'=h}^{H} Y_{h'} \mid s\right]. \qquad (9)$$

This condition ensures that the potential function upper bounds the optimal state value while being informative by not exceeding $H$ (assuming $Y_h \leq 1$).

Next, we will show that this is indeed a sufficient condition for guaranteed sample efficiency improvement under reward

---

[4] A function $f(n) = o(g(n))$ if for all $c > 0$, there exists $k > 0$ such that $f(n) < cg(n)$ for all $n \geq k$.

shaping. We start with writing the learned Q-value under shaping at step $h$ episode $k$ following Algo. 1 as follows,

$$Q_h^k(s, x) =$$
$$\sum_{i=1}^{t} \alpha_t^i \left( y_h - \phi_h(s) + (\hat{\mathbb{P}}_h^{k_i} V_{h+1}^{k_i} + \hat{\mathbb{P}}_h^{k_i} \phi_{h+1})(s, x) + b_i \right),$$

where we use $\hat{\mathbb{P}}_h^{k_i}$ to denote the empirical transition of episode $k_i$. That is, for the value of a function mapping from state space to real numbers, $f : \mathcal{S} \mapsto \mathbb{R}$, its value given the next state $s'$ in episode $k_i$ as input is denoted as, $f(s') = (\hat{\mathbb{P}}_h^{k_i} f)(s, x)$. Similarly, we write the expected value of such function $f$ over the transitions of an online CMDP under reward shaping, $\mathcal{M}'$, as $(\mathbb{P}_h f)(s, x) = \mathbb{E}_{\mathcal{M}'_\pi}[f(s')|s, x]$. $\alpha_t^i$ is the cumulative learning rates defined as $\alpha_t^0 = \Pi_{j=1}^{t}(1 - \alpha_j)$ and $\alpha_t^i = \alpha_i \Pi_{j=i+1}^{t}(1 - \alpha_j)$. Note that we also assume deterministic reward functions for notation wise simplicity (Jin et al., 2018). With the notations above, we bound the difference between the learned Q-value and the optimal Q-value with high probability in both direction.

**Lemma 4.2** (Bounded Differences Between $Q_h^k$ and $Q_h^*$). *With probability at least $1 - 2p$, the difference between the learned Q-value at the beginning of episode $k$ and step $h$ and the optimal Q-value can be bounded as $0 \leq Q_h^k(s, x) - Q_h^*(s, x) \leq \alpha_t^0(-Q_h^*(s, x)) + \sum_{i=1}^{t} \alpha_t^i \left[ (\hat{\mathbb{P}}_h^{k_i} V_{h+1}^{k_i} - \hat{\mathbb{P}}_h^{k_i} V_{h+1}^*)(s, x) \right] + 3b_t$ where $b_t$ is the UCB bonus as in Algo. 1.*

The final expected regret bound can be built upon this bounded Q-value difference via a gap-dependent decomposition. We can then bridge this gap dependent regret decomposition with the difference between learned Q-values and optimal Q-values on non-optimal actions based on the fact that $Q_h^k(s, x) - Q_h^*(s, x) \geq \Delta_h(s, x)$ (Lem. H.7),

$$\mathbb{E}\left[\text{Regret}(K)\right] = \mathbb{E}_{\mathcal{M}'}\left[\sum_{k=1}^{K}\sum_{h=1}^{H} \Delta_h(s_h^k, x_h^k)\right]$$
$$\leq (1 - 2p)\sum_{h=1}^{H}\sum_{k=1}^{K}\left(Q_h^k(s_h^k, x_h^k) - Q_h^*(s_h^k, x_h^k)\right) + 2pTH.$$

Utilizing the function $\text{clip}[x|\delta] = x \cdot \mathbb{1}[x \geq \delta], x, \delta \in \mathbb{R}$ introduced by Simchowitz & Jamieson (2019) and its properties (Lem. G.2, Lem. G.3), we can further write the difference between learned Q-values and optimal Q-values as a recursion,

$$Q_h^k(s_h^k, x_h^k) - Q_h^*(s_h^k, x_h^k) \leq \text{clip}\left[3b_t \left| \frac{\Delta(s_h^k, x_h^k)}{2H} \right.\right]$$
$$+ (1 + \frac{1}{H})\sum_{i=1}^{t} \alpha_t^i \left[ (Q_{h+1}^{k_i} - Q_{h+1}^*)(s_{h+1}^{k_i}, x_{h+1}^{k_i}) \right],$$

where we unify the optimality gaps of each state action pairs across time steps via $\Delta(s, x) = \min_h \Delta_h(s, x) =$

**Algorithm 1** Q-UCB Shaping

1: **Input:** Potential function $\phi_h(s), \forall s, h$. Const. $c > 0$.
2: $Q_h^1(s, x) = 0, N_h(s, x) = 0, \forall (s, x, h) \in \mathcal{S} \times \mathcal{X} \times [H]$.
3: Calculate maximum potential, $\phi_m = \max_s \phi(s)$.
4: **for** $k = 1$ **to** $K$ **do**
5:     Observe initial state $s_1$.
6:     **for** $h = 1$ **to** $H$ **do**
7:         Take action $x_h = \arg \max_x Q_h^k(s_h, x)$.
8:         Observe $s_{h+1}, y_h$.
9:         Update visitation counter and UCB bonus, $t = N_h(s_h, x_h) \leftarrow N_h(s_h, x_h) + 1, b_t = c\sqrt{H\phi_m^2\iota/t}$
10:         Calculate shaped reward,
        $y_h' = y_h + \phi_{h+1}(s_{h+1}) - \phi_h(s_h)$.
11:         Update Q-value,
        $Q_h^{k+1}(s_h, x_h) = (1 - \alpha_t)Q_h^k(s_h, x_h) + \alpha_t(y_h' + V_{h+1}^k(s_{h+1}) + b_t)$.
12:         Update value function,
        $V_h^{k+1}(s_h) = \min\{\phi_h(s_h), \max_x Q_h^{k+1}(s_h, x)\}$.
13:     **end for**
14: **end for**

---

$\min_h V_h^*(s) - Q_h^*(s, x)$ where $x$ is a suboptimal action. Though we can already solve the recursion and calculate the regret based on the property of the $\text{clip}(\cdot)$ function (Lem. G.3), a direct summation cannot reveal the benefits of reward shaping to the online learner. Inspired by Gupta et al. (2022), we denote the set of state-action pairs that are far from being even the second best choices as pseudo-suboptimal state-action pairs based on the minimum optimal gap, $\Delta(s, x)$.

**Definition 4.3** (Pseudo-Suboptimal State-Action Pairs)**.** We define the set of pseudo-suboptimal state-action pairs as,

$$\text{Sub}_\Delta = \{(s, x) \in \mathcal{S} \times \mathcal{X} \mid \exists h, \delta_h(s, x) \leq V_h^*(s)\}, \quad (10)$$

where $\delta_h(s, x) = y - \phi_h(s) + 2\mathbb{E}[\phi_{h+1}(s') \mid s, x] + \Delta(s, x)$.

With this set of pseudo-suboptimal state-action pairs, we show that the total number of visits made to such pairs during training can be bounded, contributing to a reduction on our final regret bound in Thm. 4.5.

**Lemma 4.4** (Bounded Number of Visits to $\text{Sub}_\Delta$)**.** *The number of visits to* $(s, x) \in \text{Sub}_\Delta$, $t = N_h^k(s, x)$, *is bounded by,* $t \leq \frac{16c^2 H \phi_m^2 \iota}{\Delta^2(s, x)}$.

By treating the summation of clipped terms in the regret differently with respect to whether the state action pairs belong to the set $\text{Sub}_\Delta$ or not, we arrive at a two-part regret bound that both demonstrates the efficiency gain of reward shaping and subsumes prior result on the gap-dependent regret of Q-UCB (Yang et al., 2021).

**Theorem 4.5** (Regret Bound of Algo. 1)**.** *Given a potential function $\phi_h(\cdot)$, with its maximum value defined as*

$\phi_m = \max_{s,h} \phi_h(s)$, *after running algorithm Algo. 1 for $K$ episodes of length $H$, the expected regret is bounded by,*

$$\mathcal{O}\left( \sum_{s,x \in \text{Sub}_\Delta} \frac{H^5 \log(SAT)}{\Delta(s, x)} + \sum_{s,x \notin \text{Sub}_\Delta} \frac{H^6 \log(SAT)}{\Delta(s, x)} \right).$$

*where $T = KH$ is totally number of steps; $\text{Sub}_\Delta$ is the set of pseudo-suboptimal state action pairs and $\Delta(s, x) = \min_h \Delta_h(s, x)$, for all $h \in [H]$.*

See App. H.3 for the full version regret bound and proof details. The first part of the regret is for the set of state-action pairs on which offline learned shaping functions would cut unnecessary over explorations resulting in an order of magnitude better dependence on the horizon factor $H$. This improved dependence on $H$ even matches with state-of-the-art Q-learning variant in the literature (Xu et al., 2021). The second part corresponds to the set of state-action pairs that our shaping function cannot decide when to stop exploring during learning matching the gap-dependent regret bound of the vanilla Q-UCB (Yang et al., 2021).

## 5. Experiments

In this section, we show simulation results verifying that: **(1)** Q-UCB with our proposed shaping function enjoys better sample efficiency , and **(2)** the policy learned by our shaping pipeline at convergence is the optimal policy for an interventional agent. The baselines include vanilla Q-UCB (No Shaping), Q-UCB shaping with minimum state values learned from offline datasets (Shaping + Min Beh. Value), shaping with maximum offline state values (Shaping + Max Beh. Value), and shaping with average offline state values (Shaping + Avg Beh. Value). We test those algorithms in a series of customized windy MiniGrid environments (Zhang & Bareinboim, 2024; Chevalier-Boisvert et al., 2018).

We use tabular state-action representations and the rewards are all deterministic in those environments. There is a step penalty of $-0.1$, $+0.2$ for getting a coin, $0$ for reaching the goal, and $-1$ for touching the lava. The episode ends immediately when either a goal/lava grid is reached or the episode length hits the horizon limit. In windy grid worlds, the state transition is determined by the resultant force of both agent's action and the wind direction if the agent tries to move. For example, if the agent wants to move right when there is a north wind (wind blowing from the north), the agent will end up being in the lower right grid concerning its original location instead of the right-hand side grid. However, the wind direction can only be observed by more capable behavioral agents. In our collected offline datasets, wind direction is thus a hidden confounder forming a CMDP. See App. D for detailed experiment setups and more results.

In Windy Empty World (Fig. 3a), there is a uniform wind

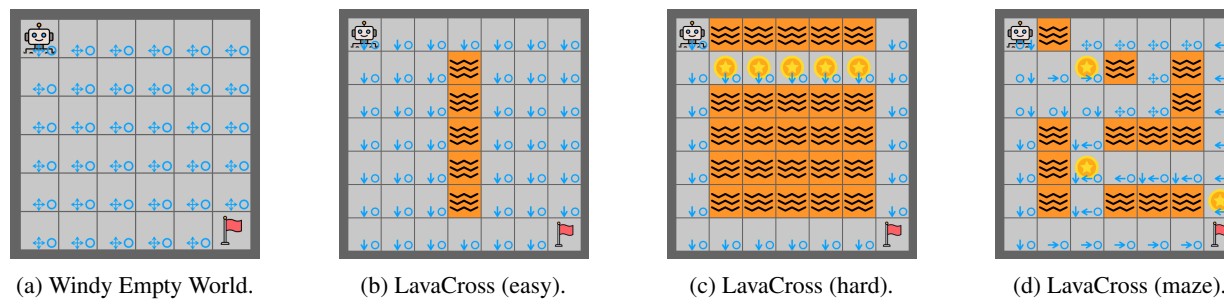

(a) Windy Empty World.      (b) LavaCross (easy).      (c) LavaCross (hard).      (d) LavaCross (maze).

Figure 3. Windy grid worlds. The blue arrows and circles in the lower right of each grid denote the possible wind directions. The flag is the goal and the orange tile is the lava. The agent's task is to reach the goal quickly without touching lava and collect coins if possible.

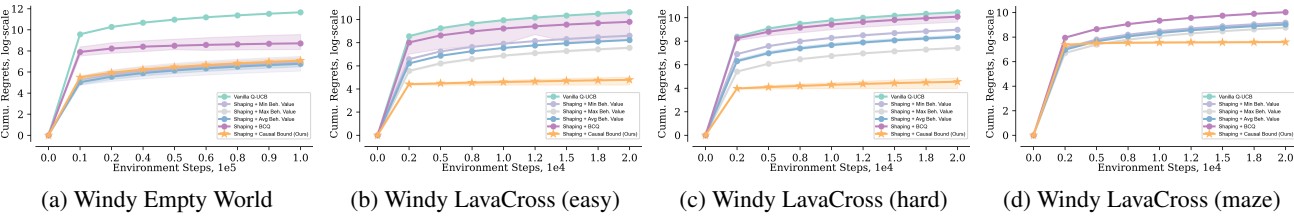

(a) Windy Empty World      (b) Windy LavaCross (easy)      (c) Windy LavaCross (hard)      (d) Windy LavaCross (maze)

Figure 4. Cumulative regrets in windy grid worlds. Lower is better. Orange curve is ours.

from each direction and a big chance for no wind (denoted as a hollow blue circle). As an interventional agent, there isn't much extra information it can utilize from the shaping function. Thus, in Fig. 4a, the cumulative regret of our method and other shaping baselines are on par with each other. Note that we do expect the vanilla Q-UCB not to perform well in all our windy grid worlds mainly because that it assumes reward being in $[0, 1]$ and initializes overly optimistically as $H$. Yet, our Q-UCB Shaping (Algo. 1) lifts this restriction and learns well with simply zero initializations.

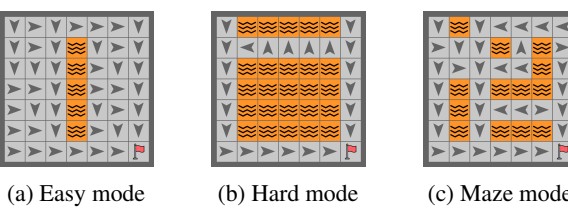

(a) Easy mode      (b) Hard mode      (c) Maze mode

Figure 5. Optimal polices learned by our proposed method.

In both Windy LavaCross easy mode and hard mode, there is a strong north wind and only a slight chance of being still across the map. We see from Fig. 3b and Fig. 3c that only the lower left L-shaped route is safe for an interventional agent. We provide offline datasets from both a conservative behavioral agent taking the safe route and an adventurous behavioral agent taking the risky northern route. The adventurous behavioral agent only moves when there is no wind. This fact is not reflected in the offline datasets though. When shaping with such values, the mixed learning signal from the adventurous agent slows down the learning process since the interventional agent cannot observe the wind and

gets punished constantly by taking the risky northern route. Only our proposed shaping with causal offline upper bound provides the correct learning signal and helps the online interventional agent converge efficiently to the right optimal policy (Fig. 5a, Fig. 5b).

In a more challenging LavaCross maze (Fig. 3d), there are three coins. Though the lower left L-shaped route is still a safe option, there is a good chance for the agent to get the middle coin given the wind distribution Fig. 4d. The other two coins are traps that only behavioral agents sensing the wind could get. If the interventional agent is shaped with such confounded values, there is a high chance that it will be pushed into lava by the wind. As a result, we see in Fig. 4d that shaping with our causal upper bound helps the agent converge and find the right optimal policy (Fig. 5c).

## 6. Conclusion

In this work we study the problem of designing potential-based reward-shaping functions automatically with confounded offline datasets from a causal perspective. Though the optimal state value is a competitive candidate for state potentials, it's not easily accessible or identifiable from confounded offline datasets. We tackle this challenge by extending the Bellman Optimal Equation to confounded MDPs to robustly upper bound the optimal state values for online interventional agents. Then, we propose a modified model-free learner, Q-UCB Shaping, which has a better regret bound than the vanilla Q-UCB when using our proposed potentials that upper bound the optimal state values. The effectiveness of our method is also verified empirically.

# Acknowledgements

This research was supported in part by the NSF, ONR, AFOSR, DoE, Amazon, JP Morgan, and the Alfred P. Sloan Foundation.

# Impact Statement

This paper presents work whose goal is to advance the field of Machine Learning. There are many potential societal consequences of our work, none of which we feel must be specifically highlighted here. One major reason is that the potential-based reward-shaping framework we considered does not alternate the optimal policy of the agent, which is an inherent property of the problem. Our proposed method solely serves as an acceleration for training such agents.

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

## A. Related Work

**Reward Shaping.** Before Ng et al. (1999) proposed the Potential-Based Reward Shaping (PBRS), the idea of transforming and modifying rewards to better facilitates learning has been studied in various problem settings (Saksida et al., 1997; Randløv & Alstrøm, 1998). Ng et al. (1999) first formalized a shaping framework that guarantees the policy invariance under reward transformation. Though this policy invariance comes at a price that shaping functions are limited to certain forms of state potential functions. There are numerous successful applications of PBRS (Brys et al., 2015; Harutyunyan et al., 2015) but there are also a growing numbers of papers that focus on using carefully designed biased shaping functions (not following the PBRS framework) (Ibrahim et al., 2024). Such shaping functions have shown effectiveness in robots playing rubic cubes (OpenAI et al., 2019), in autonomous driving (Wu et al., 2023) and more. People are also not satisfied with designing shaping functions manually and tries to learn shaping functions automatically, either serving as an exploration bonus or a supplement to the task rewards (Zou et al., 2019; Ma et al., 2024; Pathak et al., 2017; Yuan et al., 2023; Raileanu & Rocktäschel, 2020; Devidze et al., 2022).

**Off-Policy Evaluation and Learning.** Off-policy learning has a long history in RL dating back to the classic algorithms of Q-learning (Watkins, 1989; Watkins & Dayan, 1992), importance sampling (Swaminathan & Joachims, 2015; Jiang & Li, 2016), and temporal difference (Precup et al., 2000; Munos et al., 2016). Recently, people also propose to utilize offline datasets to warm start the training (Nakamoto et al., 2023; Bhargava et al., 2023; Kumar et al., 2020), augmenting online training replay buffer (Song et al., 2023; Ball et al., 2023) or incorporating imitation loss with offline data (Kang et al., 2018; Zhu et al., 2018). However, these work rely on a critical assumption that there is no unobserved confounders in the environment. While this assumption is generally true when the off-policy data is collected by an interventional agent, offline datasets generated by potentially unknown sources can easily break this assumption (Levine et al., 2020). In recent years, there is also a growing interest in identifying policy values from confounded offline data (Shi et al., 2022; Miao et al., 2022; Guo et al., 2022; Bennett & Kallus, 2024) or partially identifying the policy values via bounding (Zhang & Bareinboim, 2024; 2019).

**Regret Analysis of Finite Horizon Episodic Tabular MDPs.** Many of the prior work on regret analysis focus on the model-based setting where the transitions and reward distributions are estimated from collected data and planning is then executed in the learned model to extract the optimal policy (Singh & Yee, 1994; Kearns & Singh, 1998; Kakade, 2003; Dann & Brunskill, 2015; Dann et al., 2017; Simchowitz & Jamieson, 2019). In this work, we mainly focus on the regret analysis for model-free learners (Strehl et al., 2006; Jin et al., 2018; Wang et al., 2020; Xu et al., 2021; Chen et al., 2024). More specifically, our analysis focuses on a Q-learning variant, Q-UCB (Jin et al., 2018) that incorporates the UCB bonus into Q-learning showing model-free learners can also enjoy $\sqrt{T}$ regret as the model-based learners do. Later, Yang et al. (2021) provides a more fine-grained gap-dependent regret analysis on Q-UCB showing that it also enjoys a logarithmic dependency on total training steps $T$.

## B. Walking Robot Example Details

From the environment set up, there are two ways of walking optimally.

If the agent can observe the step size required to stabilize itself ($U_h$), it should always act as $X_h = U_h$ when $F_h = 0$. Under this perfect behavioral policy, a walking process should be: (1) When the robot is stable, $F_h = 1$, it takes a big step $X_h = 1$ and transit into an unstable status, $F_{h+1} = 0$. At this stage, the agent's location does not change, $L_{h+1} = L_h$; then (2) the agent takes an appropriate size step, $X_h = U_h$, transits back to stable status $F_{h+2} = 1$ and move forward by one step, $L_{h+2} = L_{h+1} + 1$. The whole process resembles human walking as a "controlled falling" process. This is the optimal behavioral policy. Formally, this is defined as $\beta_h^{(1)} : X_h \leftarrow U_h$ if $F_h = 0$ else $X_h \leftarrow 1$. (In Example 2, the incompetent behavioral policy is defined as $\beta_h^{(2)} : X_h \leftarrow \neg U_h$.)

However, there is an alternative approach to walking steadily when the robot cannot determine the required step size $U_t$ to stabilize itself. By taking small step size every time, $X_t = 0$, the agent always stays in stable status $F_t = 1$ and moves forward steadily in each time step. This is the optimal interventional policy. Formally, this is written as $\pi(X_t|L_t, F_t) = 0$.

To see the quantitative difference between the optimal behavioral policy and optimal interventional policy, we calculated the optimal state values as shown in the below chart. Note that for each state with given $L_t, F_t$, the optimal behavioral state values are the same for all $U_t$. We can see from the table that while the optimal behavioral policy can achieve optimal rewards from any stability status, an interventional agent should only stay in the stable status due to observational mismatch.

*Table 1.* Optimal behavioral and interventional policy values.

| $L_t$ | Behavioral | | Interventional | |
|---|---|---|---|---|
| | $F_t = 0$ | $F_t = 1$ | $F_t = 0$ | $F_t = 1$ |
| 0 | 10. | 10. | 5.0 | 5.5 |
| 1 | 9. | 9. | 4.5 | 5. |
| 2 | 8. | 8. | 4. | 4.5 |
| 3 | 7. | 7. | 3.5 | 4. |
| 4 | 6. | 6. | 3. | 3.5 |
| 5 | 5. | 5. | 2.5 | 3. |
| 6 | 4. | 4. | 2. | 2.5 |
| 7 | 3. | 3. | 1.5 | 2. |
| 8 | 2. | 2. | 1. | 1.5 |
| 9 | 1. | 1. | 0. | 1. |
| 10 | 0. | 0. | 0. | 0. |

In the meantime, state values of the behavioral policies estimated via Monte-Carlo simulation during the offline dataset collection are shown in Table 2. We see a clear trend that for the good behavioral policy $\pi_1$, it does not show any preference

*Table 2.* Confounded values of behavioral policies.

| $L_t$ | Beh. Policy $\pi_1$ | | Beh. Policy $\pi_2$ | |
|---|---|---|---|---|
| | $F_t = 0$ | $F_t = 1$ | $F_t = 0$ | $F_t = 1$ |
| 0 | 10. | 10. | -22.5 | -41.7 |
| 1 | 9. | 9. | -21.8 | -40.1 |
| 2 | 8. | 8. | -21.0 | -38.4 |
| 3 | 7. | 7. | -20.25 | -36.9 |
| 4 | 6. | 6. | -19.5 | -34.5 |
| 5 | 5. | 5. | -18.75 | -33.6 |
| 6 | 4. | 4. | -18.0 | -32.5 |
| 7 | 3. | 3. | -17.3 | -32.1 |
| 8 | 2. | 2. | -16.5 | -31.5 |
| 9 | 1. | 0. | 0. | 0. |
| 10 | 0. | 0. | 0. | 0. |

over the stability status while for a bad behavioral policy $\pi_2$, it even penalizes being stable. Thus, both confounded values are not a good approximation for our optimal interventional state value. The final state potential learned by our method, on the other hand, is shown in Table 3. We see clearly that our proposed algorithm successfully learns an upper bound to the optimal interventional state values and also showing a clear tendency towards the stable status ($F_t = 1$), the same as the optimal interventional values do.

## C. Causal Upper Bound Algorithm

We present the full pseudo-code for calculating causal upper bounds on optimal state values from multiple offline datasets in Algo. 2.

## D. Detailed Experiment Design and More Results

In this section, we provide a detailed description of experiment setups and additional experiment results.

*Table 3.* Causal state value upper bound.

| $L_t$ | $F_t = 0$ | $F_t = 1$ |
|---|---|---|
| 0 | 7.6 | 9.3 |
| 1 | 7.5 | 9.3 |
| 2 | 7.5 | 9.3 |
| 3 | 7.6 | 9.3 |
| 4 | 7.6 | 9.3 |
| 5 | 7.9 | 9.1 |
| 6 | 8.0 | 8.9 |
| 7 | 7.9 | 8.6 |
| 8 | 7.6 | 7.6 |
| 9 | 5.3 | 5.3 |
| 10 | 0. | 0. |

---

**Algorithm 2** Causal Upper Bound Potential

---

1: **Input:** Offline datasets, $\mathcal{D}^{(i)}, i = 1, ..., N$ and reward upper bound, $b$.
2: **for** $i = 1$ **to** $N$ **do**
3:    **while** Not Converged **do**
4:       **for** $h = H$ **to** 1 **do**
5:          **for** state $s \in \mathcal{S}$ **do**
6:             **for** action $x \in \mathcal{X}$ **do**
7:                **if** $s, x$ not visited in $\mathcal{D}_h^{(i)}$ **then**
8:                   Continue
9:                **end if**
10:                Calculate $\overline{Q}_h^{(i)}(s, x)$ by Eq. (8)
11:             **end for**
12:             Update $\overline{V}_h^{(i)}(s) = \max_{x \in \mathcal{D}_h^{(i)}} \overline{Q}_h^{(i)}(x|s)$
13:          **end for**
14:       **end for**
15:    **end while**
16: **end for**
17: **return** $\overline{V}_h(s) = \min_{s \in \mathcal{D}_h^{(i)}} \overline{V}_h^{(i)}(s), \forall s, h \in \mathcal{S} \times [H]$

---

### D.1. Experiment Setups

All of our experiment results are obtained from a 2021 MacBook Pro with M1 chip and 32GB memory. The cumulative regret results are averaged over three random seeds. We implement the Q-UCB in a homogeneous way that there is no step indices for state spaces.

Regarding the environment parameters, for Windy Empty World, the episode length is set to 15 while the LavaCross series has a horizon of 20. To compensate for the hard exploration situation, we allow random initial starting states over the whole map walkable area. For training steps, we set a total of 100K environment steps for Windy Empty World and 20K for the LavaCross series. We represent the wind distribution in those windy Grid World as a five-tuple meaning the probability mass for (WEST WIND, NORTH WIND, EAST WIND, SOUTH WIND, NO WIND). We list them as follows,

- Windy Empty World: $(.1, .1, .1, .1, .6)$
- Windy LavaCross (easy): $(0, .8, 0, 0, .2)$
- Windy Lava Cross (hard): $(0, .7, 0, 0, .3)$
- Windy Lava Cross (maze): $\begin{cases} (0., .5, 0., 0., .5) \text{ if } s[0] = 0 \\ (.5, 0., 0., 0., .5) \text{ if } s[1] = 7 \\ (.15, .15, .15, .15, .4) \text{ otherwise} \end{cases}$

And for each environment, we have three different behavioral policies for collecting offline datasets. Here we list their brief descriptions as below,

- **Windy Empty World**: The first one is a good behavioral policy that stands still when the wind is blowing the agent away from the goal and go towards the goal when the wind direction is in its favor. The second one is a bad behavioral policy that follows the good policy half of the time while being fully random the other half. And the last one is a fully random policy.

- **Windy LavaCross (easy)**: The first one is a good behavioral policy that takes the L-shaped safe route staying tight to the lower and left side walls. The second one is a bad behavioral policy that tries to cross the upper side gateway when there is no wind. And the last one is again a fully random policy.

- **Windy Lava Cross (hard)**: The first one is a good behavioral policy that takes the same safe route in the lower left side. The second one is a bad behavioral policy that collects those coins on the northern bank of the lava lake when there is no wind then reach the goal. And the last one is also a fully random policy.

- **Windy Lava Cross (maze)**: The first one is a good behavioral policy that goes directly to the goal location. The second one is less preferred that it takes a bit detour to get the extra coin near the goal. And the last behavioral policy goes directly to the coin on upper right corner without reaching the goal at all.

## D.2. Extra Experiment Results

Additionally, we have another LavaCross maze with fewer coins. The safe route is still the lower left L-shaped route. The behavioral agents generating offline datasets in this environment include a conservative one taking the safe route, an adventurous one taking the detour to get an extra coin before reaching the goal, and a radical one only looking for the top right corner coin without even reaching the goal. As shown in Fig. 6a, only shaping with our proposed causal upper bound helps the online interventional agent converge efficiently to the right policy (Fig. 6c,Fig. 6b). We further provide the ratio of states where the learned policy is optimal in App. D.2. Note that in the table we included one more baseline which is a deep neural network based learner, BCQ (Fujimoto et al., 2019).

*Table 4.* Optimal ratio across different environments. Higher is better.

| Algo./Env. | Windy Empty World | LavaCross(easy) | LavaCross(hard) | LavaCross(maze) |
|---|---|---|---|---|
| No Shaping Q-UCB | 0.00 | 0.00 | 0.00 | 0.03 |
| Shaping + Causal Bound(Ours) | 0.76 | **0.70** | **0.81** | **0.90** |
| Shaping + Min. Val. | 0.72 | 0.49 | 0.30 | 0.24 |
| Shaping + Max. Val. | 0.73 | 0.52 | 0.59 | 0.33 |
| Shaping + Avg. Val. | 0.74 | 0.50 | 0.49 | 0.24 |
| BCQ | 0.79 | 0.50 | 0.67 | 0.50 |
| Shaping + BCQ | **0.88** | 0.31 | 0.22 | 0.03 |

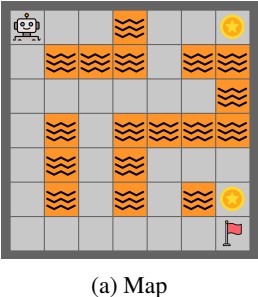

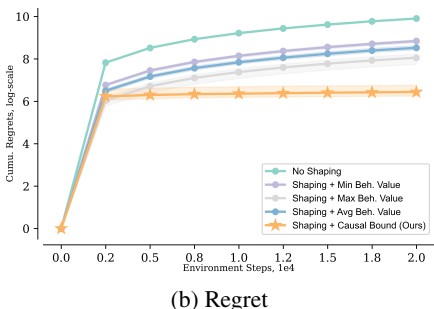

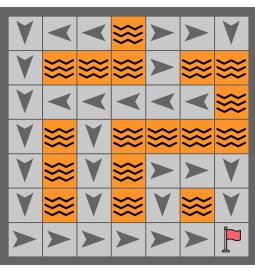

(a) Map         (b) Regret        (c) Learned Policy

*Figure 6.* LavaCross maze with fewer coins.

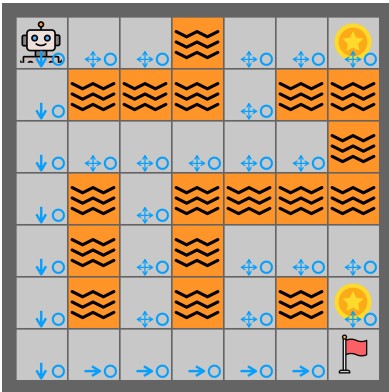

*Figure 7.* Wind distribution in the LavaCross maze with fewer coins.

# E. Shaping and Initialization

If we have the optimal state value, warm-starting the online learning by initializing state values with such optimal ones seems more straightforward than shaping. But their effects on learning are only the same under unrealistically strict conditions (Wiewiora, 2003). That is, given the same sequence of samples, the learned Q-values (Watkins & Dayan, 1992) after shaping is always less than the one learned without shaping by the amount of the potential functions. A direct implication of this result is that the policy learned under shaping at each time step is indifferent from the one learned without shaping, thus, learning efficiency is unchanged.

**Proposition E.1.** *Given a fixed sequence of samples, the policy distribution learned under shaping is equivalent to that learned without shaping but initialized with $\phi(\cdot)$.*

There is also prior work showing that for the epsilon-greedy exploration strategy, potential-based reward shaping does not directly affect learning efficiency. Instead, how optimistic the initialization is compared with the potential functions affects the learning efficiency (Dann et al., 2022). While people also find that for model-based learners, certain forms of reward shaping can boost learning efficiency provably (Gupta et al., 2022).

# F. Original Q-UCB Algorithm

We restate the pseudo-code for origianl Q-UCB in Algo. 3. For more details please see Jin et al. (2018).

---

**Algorithm 3** Q-UCB

---

1: Initialize $Q_h^1(s, x) \leftarrow H$, $N_h(s, x) = 0$, for all $(s, x, h) \in \mathcal{S} \times \mathcal{X} \times [H]$.
2: **for** $k = 1$ **to** $K$ **do**
3:    Observe initial state $s_1$.
4:    **for** $h = 1$ **to** $H$ **do**
5:       Take action $x_h = \arg\max_x Q_h^k(s_h, x)$ and observe $s_{h+1}, y_h$.
6:       Update visitation counter and calculate UCB bonus, $t = N_h(s_h, x_h) \leftarrow N_h(s_h, x_h) + 1$, $b_t = c\sqrt{H^3\iota/t}$
7:       Update Q-value, $Q_h^{k+1}(s_h, x_h) = (1 - \alpha_t)Q_h^k(s_h, x_h) + \alpha_t(y_h + V_{h+1}^k(s_{h+1}) + b_t)$.
8:       Update value function, $V_h^{k+1}(s_h) = \min\{H, \max_x Q_h^{k+1}(s_h, x)\}$.
9:    **end for**
10: **end for**

---

# G. Useful Results for Proofs

We will first restate some useful claims in the literature to be used later in our proofs for completeness.

**Lemma G.1** (Properties of Cumulative Learning Rates). *Let $\alpha_t = \frac{H+1}{H+t}$. We define $\alpha_t^0 = \Pi_{j=1}^t(1 - \alpha_j)$ and $\alpha_t^i = \alpha_i\Pi_{j=i+1}^t(1 - \alpha_j)$. The following properties hold for $\alpha_t^i$:*

(a) $\frac{1}{\sqrt{t}} \leq \sum_{i=1}^{t} \frac{\alpha_t^i}{\sqrt{i}} \leq \frac{2}{\sqrt{t}}$ for every $t \geq 1$.

(b) $\max_{i \in [t]} \alpha_t^i \leq \frac{2H}{t}$ and $\sum_{i=1}^{t} (\alpha_t^i)^2 \leq \frac{2H}{t}$ for every $t \geq 1$.

(c) $\sum_{t=i}^{\infty} \alpha_t^i = 1 + \frac{1}{H}$ for every $i > 1$.

In addition, we have (1) $\sum_{i=1}^{t} \alpha_t^i = 1$ and $\alpha_t^0 = 0$ for $t \geq 1$; (2) $\sum_{i=1}^{t} \alpha_t^i = 0$ and $\alpha_t^0 = 1$ for $t = 0$.

*Proof.* See Jin et al. (2018) App. B for details. $\qquad\square$

**Lemma G.2** (Property of the Clip Function). *Let* $\text{clip}\,[x|\delta] = x \cdot \mathbb{1}[x \geq \delta]$ *(Simchowitz & Jamieson, 2019). For any three positive numbers* $a, b, c$ *satisfying* $a + b \geq c$, *and for any* $x \in (0, 1)$, *the following holds,*

$$a + b \leq \text{clip}\left[a \,\Big|\, \frac{ac}{2}\right] + (1 + x)b \tag{11}$$

*Proof.* See Claim A.8 in App. A from Xu et al. (2021) for details. $\qquad\square$

**Lemma G.3** (Bounded Summation for Clipped Function). *The summation of a clipped function which scales proportionally to the inverse of the square root of the variable* $n$ *has the following bound:*

$$\sum_{n=1}^{\infty} \text{clip}\left[\frac{c}{\sqrt{n}} \,\Big|\, \epsilon\right] \leq \frac{4c^2}{\epsilon} \tag{12}$$

*Proof.* See Claim A.13 in App. A from Xu et al. (2021) for details. $\qquad\square$

# H. Proof Details

## H.1. Potential-Based Reward Shaping in CMDP Proof

This is for Prop. 2.2.

*Proof.* Because CMDP also enjoys the Markov property, the overall proof procedure highly resembles the original one in Ng et al. (1999). Only to note that the optimal policy invariance is proved in the online learning sense, which is between $\mathcal{M}'_\pi$ and $\mathcal{M}_\pi$. $\qquad\square$

## H.2. Causal Bellman Optimal Equation Proof

In this section, we derive the Causal Bellman Optimal Equation from the original Bellman Optimal Equation of MDPs. For stationary CMDPs, we also prove that our Causal Bellman Optimal Equation has a unique fixed point and the optimality of this unique fixed point.

**Theorem H.1** (Causal Bellman Optimal Equation (Thm. 3.1)). *For a CMDP environment* $\mathcal{M}$ *with reward* $Y_h \leq b, b \in \mathbb{R}$, *the optimal value of interventional policies,* $V_h^*(s), \forall s$, *is upper bounded by* $V^*(s) \leq \overline{V}_h(s)$ *satisfying the Causal Bellman Optimality Equation,*

$$\overline{V}_h(s) = \max_x \left[ P_h(x|s) \left( \widetilde{\mathcal{R}}_h(s, x) + \mathbb{E}_{\widetilde{\mathcal{T}}_h}[\overline{V}_{h+1}(s')] \right) + P_h(\neg x|s) \left( b + \max_{s'} \overline{V}_{h+1}(s') \right) \right] \tag{13}$$

*Proof.* Starting from the Bellman Optimal Equation, the optimal state value function is given by,

$$V_h^*(s) = \max_x R_h(s, x) + \sum_{s'} T_h(s, x, s') V_{h+1}^*(s') \tag{14}$$

Note that the actions here are done by an interventional agent, which is actually $\text{do}(x)$ in the context of a CMDP. We swap in the causal bounds for interventional reward and transition distribution,

$$V_h^*(s) \leq \max_x \widetilde{R}_h(s, x) P_h(x|s) + b P_h(\neg x|s) + \sum_{s'} \widetilde{T}_h(s, x, s') P_h(x|s) V_{h+1}^*(s') + P_h(\neg x|s) \max_{s''} V_{h+1}^*(s'') \tag{15}$$

where $\widetilde{\mathcal{R}}_h(s,x) = \mathbb{E}[Y_h|S_h = s, X_h = x]$, $\widetilde{\mathcal{T}}_h$ is shorthand for $\widetilde{\mathcal{T}}_h(s,x,s') = P(S_{h+1} = s'|S_h = s, X_h = x)$ and $P(x|s) = P_h(X_h = x|S_h = s)$ are estimated from the offline dataset. And $b$ is a known upper bound on the reward signal, $Y_h \leq b$. In this step, we upper bound the next state transition by assuming the best case that for the action not taken with probability $P_h(\neg x|s)$, the agent transits with probability 1 the best possible next state, $\max_{s''} V_{h+1}^*(s'')$.

Then after rearranging terms, we have,

$$V_h^*(s) \leq \max_x \left[ P_h(x|s) \left( \widetilde{\mathcal{R}}_h(s,x) + \sum_{s'} \widetilde{T}_h(s,x,s') V_{h+1}^*(s') \right) + P_h(\neg x|s) \left( b + \max_{s''} V_{h+1}^*(s'') \right) \right] \quad (16)$$

And optimizing the value function w.r.t this inequality gives us an upper bound on the optimal state value,

$$\overline{V}_h(s) \leq \max_x \left[ P_h(x|s) \left( \widetilde{\mathcal{R}}_h(s,x) + \sum_{s'} \widetilde{T}_h(s,x,s') \overline{V}_{h+1}(s') \right) + P_h(\neg x|s) \left( b + \max_{s''} \overline{V}_{h+1}(s'') \right) \right] \quad (17)$$

$\square$

More interestingly, we will show in Prop. H.2 that this will also converge to a unique fixed point in stationary CMDPs.

**Proposition H.2** (Convergence of Causal Bellman Optimal Equation in Stationary CMDPs). *The Causal Bellman Optimality Equation converges to a unique fixed point which is also an upper bound on the optimal interventional state values under the assumption that $P(s,x) > 0, \forall s, x$ in stationary CMDPs.*

*Proof.* We will first show that the following Causal Bellman Optimal operator is a contraction mapping with respect to a weighted max norm. The major proof technique is from Bertsekas & Tsitsiklis (1989) (Sec 4.3.2). Then by Banach's fixed-point theorem (Banach, 1922), this Causal Bellman Optimal operator has a unique fixed point and updating any initial point iteratively will converge to it. Then we will show that this unique fixed point is indeed an upper bound on the optimal interventional state value.

Let the operator $T$ be,

$$T\overline{V}(s) = \max_x \left[ P(x|s) \left( \widetilde{\mathcal{R}}(s,x) + \sum_{s'} \widetilde{T}(s,x,s') \overline{V}(s') \right) + P(\neg x|s) \left( b + \max_{s''} \overline{V}(s'') \right) \right]. \quad (18)$$

For arbitrary value bound, $\overline{V}_1, \overline{V}_2$, we can bound their difference under one step update by,

$$\left| T\overline{V}_1(s) - T\overline{V}_1(s) \right| \leq \max_x \left| P(x|s) \sum_{s'} \widetilde{T}(s,x,s') \left( \overline{V}_1(s') - \overline{V}_2(s') \right) + P(\neg x|s) \max_{s''} \left| \overline{V}_1(s'') - \overline{V}_2(s'') \right| \right|. \quad (19)$$

Note that we combine the terms together and extract $\max$ operator out because of its non-expansive property. Then we partition the state space as follows. Let $\mathcal{S}_1 = \{s_\emptyset\}$ be the set of a sink state (in episodic CMDP, we always transit into the sink state after we reach the horizon limit). For $k = 2, 3, ...$, we define,

$$\mathcal{S}_k = \{i | i \notin \bigcup_{k' \in [1, k-1]} \mathcal{S}_{k'} \wedge \min_x \max_{j \in \cup_{k' \in [1, k-1]} \mathcal{S}_{k'}} \mathcal{T}(i, x, j) > 0\}. \quad (20)$$

The set $\mathcal{S}_k$ intuitively represents the set of states that can transit to states closer to the sink state even with adversarial action picks. Let $\mathcal{S}_m$ be the last of these sets that is nonempty. We will show that all those nonempty sets cover the whole state space. Suppose the set $\mathcal{S}_\infty = \{i | i \notin \cup_{k=1}^m \mathcal{S}_k\}$ is nonempty. Then by definition, for any state $i \in \mathcal{S}_\infty$, there exists an action $x$ such that $\mathcal{T}(i, x, j) = 0, \forall j \in \cup_{k=1}^m \mathcal{S}_k$. This means that state $i$ cannot transit into the sink state with certain actions, which contradicts with our episodic setting that starting from any state-action pair, the episode will end at the horizon limit.

Then we define the max norm with respect to which we will later show that the difference between value update is a contraction mapping. We set the weight vector $w > 0$ in a way that each of the entry $w_i, i \in \mathcal{S}$ corresponds to a set $\mathcal{S}_k$ and all states inside the same set shares the same weight, that is, $w_i = y_k$ if $i \in \mathcal{S}_k$.

We will for now assume the following properties of sequence $\{y_k\}_{k=1}^m$ hold, prove $T$ is a contraction mapping, and then come back to show we can find such sequences.

$$1 = y_1 < y_2 < \cdots < y_m \tag{21}$$

$$\frac{y_m}{y_k}(1 - \epsilon\delta) + \frac{y_{k-1}}{y_k}\epsilon\delta \leq \gamma < 1, k = 2, ..., m \tag{22}$$

where $\epsilon = \min_{k \in [2,m]} \min_x \min_{i \in \mathcal{S}_k} \sum_{j \in \bigcup_{k' \in [1,k-1]} \mathcal{S}_{k'}} \mathcal{T}(i, x, j)$ is the minimal one-step transition probability that a state from later set $\mathcal{S}_k$ transfer to a state that is closer to the sink state and $\delta = \min_{s,x} P(s, x) > 0$ from the behavioral dataset,. From previous discussion and probability distribution validity property, we have $\epsilon \in (0, 1]$.

Let the initial difference between the value vectors be $||\overline{V}_1 - \overline{V}_2||_\infty^w \leq c$ where $c > 0$ is a constant. Then we have, for any state $s \in \mathcal{S}_{k(s)}$ where $k(s)$ means the index of the state set that $s$ belongs to,

$$\overline{V}_1(s) - \overline{V}_2(s) \leq cy_{k(s)}. \tag{23}$$

Now we can continue to write the value update difference as,

$$\frac{|T\overline{V}_1(s) - T\overline{V}_2(s)|}{cy_{k(s)}} \leq \frac{1}{y_{k(s)}} \max_x \left| P(x|s) \sum_{s'} \widetilde{T}(s, x, s') y_{k(s')} + P(\neg x|s) \max_{s''} y_{k(s'')} \right|. \tag{24}$$

$$= \frac{1}{y_{k(s)}} \max_x \left( P(x|s) \sum_{s'} \widetilde{T}(s, x, s') y_{k(s')} + P(\neg x|s) \max_{s''} y_{k(s'')} \right). \tag{25}$$

Given the fact that $y_k \leq y_m, \forall k \in [1, m]$,

$$\frac{|T\overline{V}_1(s) - T\overline{V}_2(s)|}{cy_{k(s)}} \leq \frac{1}{y_{k(s)}} \max_x \left( P(x|s) \sum_{s'} \widetilde{T}(s, x, s') y_{k(s')} + P(\neg x|s) y_m \right). \tag{26}$$

We can split the sum over next state $s'$ by differentiating whether it belongs to the sets closer to sink state than $s$ or the sets further away from the sink state than $s$,w2

$$\frac{|T\overline{V}_1(s) - T\overline{V}_2(s)|}{cy_{k(s)}} \leq \frac{1}{y_{k(s)}} \max_x \left( P(x|s) \sum_{s' \in \bigcup_{k' \in [1,k(s)-1]} \mathcal{S}_{k'}} \widetilde{T}(s, x, s') y_{k(s')} \right.$$
$$\left. + P(x|s) \sum_{s' \in \bigcup_{k' \in [k(s),m]} \mathcal{S}_{k'}} \widetilde{T}(s, x, s') y_{k(s')} + P(\neg x|s) y_m \right). \tag{27}$$

And by the property of $\{y_k\}_{k=1}^m$, $1 = y_1 < y_2 < \cdots < y_m$ and the fact that $\sum_{s'} \widetilde{T}(s, x, s') = 1$, we have,

$$\frac{|T\overline{V}_1(s) - T\overline{V}_2(s)|}{cy_{k(s)}} \leq \frac{1}{y_{k(s)}} \max_x \left( (y_{k(s)-1} - y_m) P(x|s) \sum_{s' \in \bigcup_{k' \in [1,k(s)-1]} \mathcal{S}_{k'}} \widetilde{T}(s, x, s') + y_m P(x|s) + P(\neg x|s) y_m \right) \tag{28}$$

$$\leq \frac{1}{y_{k(s)}} \max_x \left( (y_{k(s)-1} - y_m) P(x|s) \sum_{s' \in \bigcup_{k' \in [1,k(s)-1]} \mathcal{S}_{k'}} \widetilde{T}(s, x, s') \right) + \frac{y_m}{y_{k(s)}}. \tag{29}$$

By definition, $\epsilon$ lower bounds the transition probability of $\sum_{s' \in \bigcup_{k' \in [1,k(s)-1]} \mathcal{S}_{k'}} \widetilde{T}(s, x, s')$, we have,

$$\frac{|T\overline{V}_1(s) - T\overline{V}_2(s)|}{cy_{k(s)}} \leq \frac{1}{y_{k(s)}} \max_x P(x|s)(y_{k(s)-1} - y_m)\epsilon + \frac{y_m}{y_{k(s)}} \tag{30}$$

$$\leq \frac{y_m}{y_k}(1 - \epsilon\delta) + \frac{y_{k-1}}{y_k}\epsilon\delta \tag{31}$$

$$\leq \gamma \tag{32}$$

Thus, for the state value bound vectors, we have,

$$||T\overline{V}_1 - T\overline{V}_2||_\infty^w \le \gamma c \tag{33}$$

for all $\overline{V}_1, \overline{V}_2$ satisfying $||\overline{V}_1 - \overline{V}_2|| \le c$. Thus, $T$ is a contraction mapping with respect to a weighted max norm.

Lastly, we will show that a sequence $\{y_k\}_{k=1}^m$ satisfying Eq. (21) and Eq. (22) is realizable. Let $y_0 = 0, y_1 = 1$, and suppose that $y_1, y_2, ..., y_k$ have been chosen. If $\epsilon\delta = 1$, we set $y_{k+1} = y_k + 1$. If $\epsilon\delta < 1$, we set

$$y_{k+1} = 1/2(y_k + z_k) \tag{34}$$

where

$$z_k = \min_{i \in [1,k]} \left[ y_i + \frac{\epsilon\delta}{1 - \epsilon\delta}(y_i - y_{i-1}) \right]. \tag{35}$$

And we can write $z_k$ recursively as,

$$z_k = \min\{z_{k-1}, y_k + \frac{\epsilon\delta}{1 - \epsilon\delta}(y_k - y_{k-1})\}. \tag{36}$$

We will first show by induction that this sequence satisfies Eq. (21).

(1) Base Case: By definition, $y_0 < y_1$.

(2) Induction Hypothesis: Assume $y_{k-1} < y_k$ holds.

(3) Inductive Step: Our goal is to show that $y_k < y_{k+1}$. By definition, $y_{k+1} = \frac{1}{2}(y_k + z_k)$, the problem is reduced to whether $y_{k+1} - y_k = z_k - y_k > 0$. Since $y_k = \frac{1}{2}(y_{k-1} + z_{k-1})$ and we have $y_{k-1} < y_k$. Thus, we have $z_{k-1} > y_k$. And clearly we have $y_k + \frac{\epsilon}{1-\epsilon}(y_k - y_{k-1}) > y_k$. By the recursive update rule, we have,

$$z_k = \min\{z_{k-1}, y_k + \frac{\epsilon}{1 - \epsilon}(y_k - y_{k-1})\} \tag{37}$$

As we have shown that both term are bigger than $y_k$, we have $z_k > y_k$. Thus, the sequence $\{y_k\}_{k=1}^m$ satisfies Eq. (21).

For Eq. (22), we first notice that since $z_{m-1} - y_m = y_m - y_{m-1} > 0$, we have,

$$z_m = \min\{z_{m-1}, y_m + \frac{\epsilon}{1 - \epsilon}(y_m - y_{m-1})\} > y_m. \tag{38}$$

By definition, $z_m$ is also calculated as,

$$z_m = \min_{i \in [1,m]} \left[ y_i + \frac{\epsilon\delta}{1 - \epsilon\delta}(y_i - y_{i-1}) \right]. \tag{39}$$

Swapping this definition into Eq. (38), we have,

$$y_k + \frac{\epsilon\delta}{1 - \epsilon\delta}(y_k - y_{k-1}) > y_m \tag{40}$$

$$\frac{y_m}{y_k} < 1 + \frac{\epsilon\delta}{1 - \epsilon\delta}(1 - \frac{y_{k-1}}{y_k}) \tag{41}$$

$$(1 - \epsilon\delta)\frac{y_m}{y_k} + \epsilon\delta\frac{y_{k-1}}{y_k} < 1. \tag{42}$$

Thus, we have shown that a weight sequence $\{y_k\}_{k=1}^m$ satisfying Eq. (21) and Eq. (22) is indeed realizable.

Thus, $T$ is a contraction mapping with respect to a realizable max norm. There exists a unique fixed point $\overline{V}^*$ when we optimize $\overline{V}$ with $T$ iteratively till convergence. For the optimal state value vector $V^*$, we can also apply $T$ iteratively until it converges to the fixed point $\overline{V}^*$. By the update rule of $T$ (Eq. (17)), $\forall V, V \le TV$. Thus, we have $V^* \le \lim_{k \to \infty} T^k V^* = \overline{V}^*$ where $T^k$ denotes applying $T$ iteratively for $k$ times. The fixed point of the Causal Bellman Optimal Equation is indeed an upper bound of the optimal state value vector. $\square$

## H.3. Q-UCB with Shaping Regret Analysis Details

We define the adaptive learning rate as $\alpha_t = \frac{H+1}{H+t}$ and a shorthand notion $\iota = \log\left(|\mathcal{S}||\mathcal{X}|T/p\right)$ where $T = KH$. We also have $\phi(s_{H+1}) = 0$ according to Prop. 2.2 and assume deterministic reward functions. Throughout the proof, we also assume conservative optimism condition is satisfied (Def. 4.1).

We first present lemmas used in proving the regret bound of Algo. 1.

**Lemma H.3** (Concentration of Transition Weighted Bounded Functions). *Let $f$ be a function mapping from state space to a convex set of real values, $\mathcal{S} \mapsto [0, \phi_m], \phi_m \in \mathbb{R}$. With high probability, the following sum is bounded for all $(s, x, h, k) \in \mathcal{S} \times \mathcal{X} \times [H] \times [K]$,*

$$\forall t \in [K], \sum_{i=1}^{t} \alpha_t^i \cdot \mathbb{1}[k_i < K] \cdot (\hat{\mathbb{P}}_h^{k_i} f - \mathbb{P}_h f)(s, x) \leq \frac{b_t}{2}, \tag{43}$$

*where $t = N_h^k(s, x)$ is the visitation count at the beginning of $k$-th episode and we define the index of the episode when state-action pair $(s, x)$ is visited for the $i$-th time to be,*

$$k_i = \min\left(\{k \in [K] | k > k_{i-1} \wedge (s_h^k, x_h^k) = (s, x)\} \cup \{K + 1\}\right). \tag{44}$$

*Proof.* Note that $k_i = K + 1$ if $(s, x)$ is not visited for the $i$-th time. The sequence $\{\alpha_t^i \cdot \mathbb{1}[k_i < K] \cdot (\hat{\mathbb{P}}_h^{k_i} f - \mathbb{P}_h f)(s, x)\}_{i=0}^{t}$ is a martingale difference sequence w.r.t filtration $\mathcal{F}_{i \geq 0}$ and we have $|\alpha_t^i \cdot \mathbb{1}[k_i < K] \cdot (\hat{\mathbb{P}}_h^{k_i} f - \mathbb{P}_h f)(s, x)| \leq \alpha_i^t \cdot \phi_m$ and $\alpha_t^0 \cdot \mathbb{1}[k_0 < K] \cdot (\hat{\mathbb{P}}_h^{k_0} f - \mathbb{P}_h f)(s, x) = 0$ since no visitation occurs at all when $i = 0$. By Azuma-Hoeffding inequality, we have that $\forall \epsilon > 0$,

$$P(|\sum_{i=0}^{t} \alpha_t^i \cdot \mathbb{1}[k_i < K] \cdot (\hat{\mathbb{P}}_h^{k_i} f - \mathbb{P}_h f)(s, x)| \geq \epsilon) \leq 2 \exp\left(\frac{-\epsilon^2}{2 \sum_{i=0}^{t} (\alpha_t^i)^2 \phi_m^2}\right). \tag{45}$$

Thus, with probability $p' \leq \frac{p}{SAT}$, $|\sum_{i=0}^{t} \alpha_t^i \cdot \mathbb{1}[k_i < K] \cdot (\hat{\mathbb{P}}_h^{k_i} f - \mathbb{P}_h f)(s, x)| \geq \frac{c}{2} \sqrt{\phi_m^2 \log(SAT/p) H / t} = \frac{b_t}{2}$ where $c > 0$ is a constant. By union bound over all $(s, x, h, k) \in \mathcal{S} \times \mathcal{X} \times [H] \times [K]$, we conclude the proof for the claim. $\square$

**Lemma H.4** (Bounded Differences Between $Q_h^k$ and $Q_h^*$ (Lem. 4.2)). *The difference between the learned Q-value at the beginning of episode $k$ and step $h$ and the optimal Q-value can be bounded with high probability as follows,*

$$0 \leq Q_h^k(s, x) - Q_h^*(s, x) \leq \alpha_t^0(-Q_h^*(s, x)) + \sum_{i=1}^{t} \alpha_t^i \left[(\hat{\mathbb{P}}_h^{k_i} V_{h+1}^{k_i} - \hat{\mathbb{P}}_h^{k_i} V_{h+1}^*)(s, x)\right] + 3b_t, \tag{46}$$

*where $\mathbb{P}_h$ denotes the expected transition w.r.t the true confounded MDP and $\hat{\mathbb{P}}_h^k$ denotes the sample transition experienced by the agent at episode $k$ step $h$.*

*Proof.* First, we can rewrite the optimal Q-value as follows by the Bellman equation and Prop. 2.2,

$$Q_h^*(s, x) = \alpha_t^0 Q_h^*(s, x) + \sum_{i=1}^{t} \alpha_t^i (y_h(s, x) - \phi(s) + (\mathbb{P}_h V_{h+1}^* + \mathbb{P}_h \phi)(s, x). \tag{47}$$

The learned Q-value can be written as follows using the defined accumulative learning rates Lem. G.1,

$$Q_h^k(s, x) = \sum_{i=1}^{t} \alpha_t^i \left(y_h - \phi(s) + (\hat{\mathbb{P}}_h^{k_i} V_{h+1}^{k_i} + \hat{\mathbb{P}}_h^{k_i} \phi)(s, x) + b_i\right). \tag{48}$$

Subtracting Eq. (47) from Eq. (48), we have,

$$Q_h^k(s, x) - Q_h^*(s, x) \tag{49}$$

$$= \alpha_t^0(-Q_h^*(s, x)) + \sum_{i=1}^{t} \alpha_t^i \left((\hat{\mathbb{P}}_h^{k_i} V_{h+1}^{k_i} - \mathbb{P}_h V_{h+1}^*)(s, x) + (\hat{\mathbb{P}}_h^{k_i} \phi - \mathbb{P}_h \phi)(s, x) + b_i\right) \tag{50}$$

$$= \alpha_t^0(-Q_h^*(s, x)) + \sum_{i=1}^{t} \alpha_t^i \left((\hat{\mathbb{P}}_h^{k_i} V_{h+1}^{k_i} - \hat{\mathbb{P}}_h^{k_i} V_{h+1}^*)(s, x) + (\hat{\mathbb{P}}_h^{k_i} V_{h+1}^* - \mathbb{P}_h V_{h+1}^*)(s, x) + (\hat{\mathbb{P}}_h^{k_i} \phi - \mathbb{P}_h \phi)(s, x) + b_i\right). \tag{51}$$

By Lem. H.3, we have that with probability at least $1 - 2p$ for all $(s, x, h, k) \in \mathcal{S} \times \mathcal{X} \times [H] \times [K]$,

$$Q_h^k(s, x) - Q_h^*(s, x) \le \alpha_t^0(-Q_h^*(s, x)) + \sum_{i=1}^{t} \alpha_t^i \left( (\hat{\mathbb{P}}_h^{k_i} V_{h+1}^{k_i} - \hat{\mathbb{P}}_h^{k_i} V_{h+1}^*)(s, x) + b_i \right) + b_t. \tag{52}$$

By Lem. G.1, we have,

$$Q_h^k(s, x) - Q_h^*(s, x) \le \alpha_t^0(-Q_h^*(s, x)) + \sum_{i=1}^{t} \alpha_t^i \left[ (\hat{\mathbb{P}}_h^{k_i} V_{h+1}^{k_i} - \hat{\mathbb{P}}_h^{k_i} V_{h+1}^*)(s, x) \right] + c\sqrt{H\phi_m^2 \iota} \cdot \sum_{i=1}^{t} \frac{\alpha_t^i}{\sqrt{i}} + b_t \tag{53}$$

$$\le \alpha_t^0(-Q_h^*(s, x)) + \sum_{i=1}^{t} \alpha_t^i \left[ (\hat{\mathbb{P}}_h^{k_i} V_{h+1}^{k_i} - \hat{\mathbb{P}}_h^{k_i} V_{h+1}^*)(s, x) \right] + 3b_t. \tag{54}$$

This concludes the proof for the right hand side. Then we will show that $Q_h^k(s, x) - Q_h^*(s, x) \ge 0$ by induction on the steps of the episode from $H$ to 1. Without losing generality, we assume $t > 0$ since when $t = 0$ it's straightforward to show this holds.

(1) Base case: When $h = H$ and for all $k$, it's the last time step and all future state values are zero by definition. We have,

$$Q_h^k(s, x) - Q_h^*(s, x) = \sum_{i=1}^{t} \alpha_t^i \left( (\hat{\mathbb{P}}_h^{k_i} \phi - \mathbb{P}_h \phi)(s, x) + b_i \right). \tag{55}$$

By Lem. H.3 and together with the discussion above for the other side, we know $(\hat{\mathbb{P}}_h^{k_i} \phi - \mathbb{P}_h \phi)(s, x) \ge -\frac{b_t}{2}$ with high probability. While by Lem. G.1, $\sum_{i=1}^{t} \alpha_t^i b_i \ge b_t$, thus, we have,

$$Q_h^k(s, x) - Q_h^*(s, x) \ge b_t + (-\frac{b_t}{2}) \ge 0. \tag{56}$$

The base case holds.

(2) Induction hypothesis: for $h = h'$ and for all $k$, $Q_h^k(s, x) - Q_h^*(s, x) \ge 0$.

(3) Induction step: Now we show that for $h = h' - 1$, the claim still holds. Recall that we can write the Q-value differences as follows when $t > 0$,

$$Q_h^k(s, x) - Q_h^*(s, x) \tag{57}$$

$$= \sum_{i=1}^{t} \alpha_t^i \left( (\hat{\mathbb{P}}_h^{k_i} V_{h+1}^{k_i} - \hat{\mathbb{P}}_h^{k_i} V_{h+1}^*)(s, x) + (\hat{\mathbb{P}}_h^{k_i} V_{h+1}^* - \mathbb{P}_h V_{h+1}^*)(s, x) + (\hat{\mathbb{P}}_h^{k_i} \phi - \mathbb{P}_h \phi)(s, x) + b_i \right). \tag{58}$$

And by Lem. G.1 and similar discussions as in the base case, we are left with terms,

$$Q_h^k(s, x) - Q_h^*(s, x) \ge \sum_{i=1}^{t} \alpha_t^i (\hat{\mathbb{P}}_h^{k_i} V_{h+1}^{k_i} - \hat{\mathbb{P}}_h^{k_i} V_{h+1}^*)(s, x). \tag{59}$$

By Algo. 1, $V_{h+1}^{k+1}(s_{h+1}) = \min\{\phi(s_{h+1}), \max_x Q_{h+1}^{k+1}(s_{h+1}, x)\}$. Thus, no matter which value $V_{h+1}^{k+1}(s_{h+1})$ is getting, by the potential function property ($\phi \ge V^*$) and the induction hypothesis, $V_{h+1}^{k+1}(s_{h+1}) \ge V_{h+1}^*(s_{h+1})$ holds.

Thus, we have concluded the proof for the left hand side. □

**Definition H.5** (Psudo-Suboptimal State-Action Pairs (Def. 4.3))**.** We define the set of pseudo-suboptimal state-action pairs to be the set that satisfies,

$$\text{Sub}_\Delta = \{(s, x) \in \mathcal{S} \times \mathcal{X} | \exists h \in [H], y_h(s, x) - \phi_h(s) + 2(\mathbb{P}_h \phi_{h+1})(s, x) + \Delta(s, x) \le V_h^*(s)\}, \tag{60}$$

where $\Delta(s, x) = \min_h \Delta_h(s, x) = \min_h(V_h^*(s) - Q_h^*(s, x))$.

**Lemma H.6** (Bounded Number of Visits to $\mathrm{Sub}_\Delta$ (Lem. 4.4)). *The number of visits to $(s,x) \in \mathrm{Sub}_\Delta$, $t = N_h^k(s,x)$, is bounded by,*

$$t \leq \frac{16c^2 H \phi_m^2 \iota}{\Delta^2(s,x)}. \tag{61}$$

*Proof.* When $t$ exceeds the bound,

$$t > \frac{16c^2 H \phi_m^2 \iota}{\Delta^2(s,x)}, \tag{62}$$

by property in Lem. G.1, we have,

$$2c\sqrt{H\phi_m^2\iota} \sum_{i=1}^{t} \frac{\alpha_t^i}{\sqrt{i}} \leq 2c\sqrt{H\phi_m^2\iota}\frac{2}{\sqrt{t}} < \Delta(s,x). \tag{63}$$

This implies $2\sum_{i=1}^{t} \alpha_t^i b_i < \Delta(s,x)$. And by Lem. H.3, with high probability, we have,

$$\sum_{i=1}^{t} \alpha_t^i \cdot \hat{\mathbb{P}}_h^{k_i} f \leq \sum_{i=1}^{t} \alpha_t^i \cdot \mathbb{P}_h f + \frac{b_t}{2} \leq \sum_{i=1}^{t} \left( \alpha_t^i \cdot \mathbb{P}_h f + \frac{b_i}{2} \right) \tag{64}$$

where $f$ is a bounded real value function $f : \mathcal{S} \mapsto [0, \phi_m]$. Apply this to the learned Q value $Q_h^k$ and by the conservative optimism condition (Def. 4.1) that $V_h^{k_i} \leq \phi$,

$$Q_h^k(s,x) = \sum_{i=1}^{t} \alpha_t^i \left( y_h - \phi_h(s) + (\hat{\mathbb{P}}_h^{k_i} V_{h+1}^{k_i} + \hat{\mathbb{P}}_h^{k_i} \phi_{h+1})(s,x) + b_i \right) \tag{65}$$

$$\leq \sum_{i=1}^{t} \alpha_t^i (y_h - \phi_h(s) + 2(\mathbb{P}_h \phi_{h+1})(s,x) + 2b_i). \tag{66}$$

$$\tag{67}$$

Plug $2\sum_{i=1}^{t} \alpha_t^i b_i < \Delta(s,x)$ into this,

$$Q_h^k(s,x) < \sum_{i=1}^{t} \alpha_t^i (y_h - \phi_h(s) + 2(\mathbb{P}_h \phi_{h+1})(s,x) + \Delta(s,x)). \tag{68}$$

For $(s,x) \in \mathrm{Sub}_\Delta$, by the definition of $\mathrm{Sub}_\Delta$ and the optimistic property of the learned Q-value (Lem. H.4), we have,

$$Q_h^k(s,x) < \sum_{i=1}^{t} \alpha_t^i \cdot V_h^*(s) \tag{69}$$

$$= Q_h^*(s,x^*) \tag{70}$$

$$\leq Q_h^k(s,x^*). \tag{71}$$

This indicates that when the number of visits to the state action pairs in $\mathrm{Sub}_\Delta$, $t$, exceeds the proposed bound, its learned Q value will be upper bounded by the learned Q value of the optimal actions, and by the greedy action selection rule in Algo. 1, such actions will no longer be chosen going forward. □

**Lemma H.7** (Lower Bound on Sub-optimal Action's Q-Value Differences). *For actions that are suboptimal but being selected in the algorithm, $x \neq x^*$, with high probability, we have,*

$$Q_h^k(s,x) - Q_h^*(s,x) \geq \Delta_h(s,x) \tag{72}$$

*Proof.* Because such actions are selected over the optimal, their learned Q-value must be larger. And by Lem. H.4, $Q_h^k(s, x^*) \geq Q_h^*(s, x^*)$ with high probability. Thus, we have,

$$Q_h^k(s, x) - Q_h^*(s, x) \geq Q_h^k(s, x^*) - Q_h^*(s, x) \tag{73}$$

$$\geq Q_h^*(s, x^*) - Q_h^*(s, x) \tag{74}$$

$$= V_h^*(s) - Q_h^*(s, x) \tag{75}$$

$$= \Delta_h(s, x). \tag{76}$$

$\square$

Now we are ready to prove the main theorem.

**Theorem H.8** (Regret Bound for Algo. 1. (Thm. 4.5)). *Given a potential function $\phi(\cdot)$, with its maximum value defined as $\phi_m$, after running algorithm Algo. 1 for $K$ episodes with $H$ steps each, the expected regret is bounded by,*

$$\tilde{\mathcal{O}} \left( \sum_{s, x \in \mathrm{Sub}_\Delta} \frac{H^3 \phi_m^2}{\Delta(s, x)} + \sum_{s, x \notin \mathrm{Sub}_\Delta} \frac{H^4 \phi_m^2}{\Delta(s, x)} \right), \tag{77}$$

*where $\mathrm{Sub}_\Delta$ is the set of pseudo suboptimal state action pairs and $\Delta(s, x) = \min_h \Delta_h(s, x)$, for all $h \in [H]$.*

*Proof.* By the definition of expected regret over trajectories $\tau$, we can decompose it as follows,

$$\mathbb{E}\left[\mathrm{Regret}(K)\right] = \mathbb{E}_\tau \left[ \sum_{k=1}^K V_1^*(s_1^k) - V_1^{\pi_k}(s_1^k) \right] \tag{78}$$

$$= \mathbb{E}_\tau \left[ \sum_{k=1}^K V_1^*(s_1^k) - Q_1^*(s_1^k, x_1^k) + Q_1^*(s_1^k, x_1^k) - V_1^{\pi_k}(s_1^k) \right] \tag{79}$$

$$= \mathbb{E}_\tau \left[ \sum_{k=1}^K \Delta_1(s_1^k, x_1^k) + \mathbb{E}_{s_2}[V_2^*(s_2) - V_2^{\pi_k}(s_2) | \pi_k, x_1^k, s_1^k] \right] \tag{80}$$

$$= \ldots \tag{81}$$

$$= \mathbb{E}_\tau \left[ \sum_{k=1}^K \sum_{h=1}^H \Delta_h(s_h^k, x_h^k) \mid \pi^k \right]. \tag{82}$$

We can split trajectories by whether the high probability event in Lem. H.4 happens. When Lem. H.4 happens with at least $1 - 2p$ probability, Lem. H.7 is also satisfied and we have,

$$\mathbb{E}\left[\mathrm{Regret}(K)\right] = \mathbb{E}_\tau \left[ \sum_{k=1}^K \sum_{h=1}^H \Delta_h(s_h^k, x_h^k) \mid \pi^k \right] \tag{83}$$

$$\leq (1 - 2p) \sum_{h=1}^H \sum_{k=1}^K \left( Q_h^k(s_h^k, x_h^k) - Q_h^*(s_h^k, x_h^k) \right) + 2pTH. \tag{84}$$

We can set failure probability to be $p = \frac{1}{2T}$ and by L.H.S of the inequality in Lem. H.4,

$$\mathbb{E}\left[\mathrm{Regret}(K)\right] \leq \sum_{h=1}^H \sum_{k=1}^K \left( Q_h^k(s_h^k, x_h^k) - Q_h^*(s_h^k, x_h^k) \right) + H. \tag{85}$$

Now, the regret bound boils down to the cumulative Q value differences. We revisit Lem. H.4 and expand the R.H.S with Bellman equations,

$$Q_h^k(s_h^k, x_h^k) - Q_h^*(s_h^k, x_h^k) \leq \sum_{i=1}^t \alpha_t^i \left[ (\hat{\mathbb{P}}_h^{k_i} V_{h+1}^{k_i} - \hat{\mathbb{P}}_h^{k_i} V_{h+1}^*)(s, x) \right] + 3b_t \tag{86}$$

$$= \sum_{i=1}^t \alpha_t^i \left[ (Q_{h+1}^{k_i} - Q_{h+1}^*)(s_{h+1}^{k_i}, x_{h+1}^{k_i}) \right] + 3b_t. \tag{87}$$

By Lem. G.2, Lem. H.7 and the fact that $\Delta(s,x) \le \Delta_h(s,x)$,

$$Q_h^k(s_h^k, x_h^k) - Q_h^*(s_h^k, x_h^k) \le \text{clip}\left[3b_t \left|\frac{\Delta(s_h^k, x_h^k)}{2H}\right|\right] + (1 + \frac{1}{H})\sum_{i=1}^t \alpha_t^i \left[(Q_{h+1}^{k_i} - Q_{h+1}^*)(s_{h+1}^{k_i}, x_{h+1}^{k_i})\right]. \quad (88)$$

Since $Q_{H+1}^* = Q_{H+1}^k = 0$, we can expand the above recursion and solve for value difference of all episodes at step $h$, $\sum_{k=1}^K Q_h^k(s_h^k, x_h^k) - Q_h^*(s_h^k, x_h^k)$, as follows,

$$\sum_{k=1}^K Q_h^k(s_h^k, x_h^k) - Q_h^*(s_h^k, x_h^k) \le \sum_{k',h'\ge h}(1+\frac{1}{H})^{2(h'-h)}\text{clip}\left[3b_{N_{h'}^{k'}}\left|\frac{\Delta(s_{h'}^{k'}, x_{h'}^{k'})}{2H}\right|\right] \quad (89)$$

$$\le e^2 \sum_{k,h}\text{clip}\left[3b_{N_h^k}\left|\frac{\Delta(s_h^k, x_h^k)}{2H}\right|\right]. \quad (90)$$

The term $(1+\frac{1}{H})^2$ comes from both the coefficient $(1+\frac{1}{H})$ and the fact that $\sum_{i=1}^\infty \alpha_t^i \le 1 + \frac{1}{H}$ by Lem. G.1. Then we plug this term back into our regret bound (Eq. (85)),

$$\mathbb{E}\left[\text{Regret}(K)\right] \le e^2 H \sum_{k,h}\text{clip}\left[3b_{N_h^k}\left|\frac{\Delta(s_h^k, x_h^k)}{2H}\right|\right] + H. \quad (91)$$

We can rewrite the summations regarding the state action space and split the summation by whether $(s,x) \in \text{Sub}_\Delta$ or not,

$$\mathbb{E}\left[\text{Regret}(K)\right] \le e^2 H \sum_{k,h}\text{clip}\left[3b_{N_h^k}\left|\frac{\Delta(s_h^k, x_h^k)}{2H}\right|\right] + H \quad (92)$$

$$= e^2 H \sum_{h=1}^H \sum_{(s,x)\in\text{Sub}_\Delta}\sum_{i=1}^{N_h^K(s,x)}\text{clip}\left[3b_i\left|\frac{\Delta(s,x)}{2H}\right|\right]$$

$$+ e^2 H \sum_{h=1}^H \sum_{(s,x)\notin\text{Sub}_\Delta}\sum_{i=1}^{N_h^K(s,x)}\text{clip}\left[3b_i\left|\frac{\Delta(s,x)}{2H}\right|\right] + H. \quad (93)$$

Since from Lem. H.6, we know the total number of visits for each $(s,x) \in \text{Sub}_\Delta$ is bounded, we can apply this result on $N_h^K(s,x)$ for $(s,x) \in \text{Sub}_\Delta$. And for $(s,x) \notin \text{Sub}_\Delta$, we apply Lem. G.3 directly. Then the expected regret is bounded as follows,

$$\mathbb{E}\left[\text{Regret}(K)\right] \le 3e^2 H^2 \sum_{(s,x)\in\text{Sub}_\Delta}\sum_{i=1}^{N_h^K(s,x)}b_i + e^2 H^2 \sum_{(s,x)\notin\text{Sub}_\Delta}\frac{8H\cdot 9c^2\cdot H\phi_m^2\iota}{\Delta(s,x)} + H \quad (94)$$

$$= 3ce^2 H^2\sqrt{H\phi_m^2\iota}\sum_{(s,x)\in\text{Sub}_\Delta}\sum_{i=1}^{N_h^K(s,x)}\frac{1}{\sqrt{i}} + \sum_{(s,x)\notin\text{Sub}_\Delta}\frac{72c^2e^2\phi_m^2 H^4\iota}{\Delta(s,x)} + H \quad (95)$$

$$\le 3ce^2 H^2\sqrt{H\phi_m^2\iota}\sum_{(s,x)\in\text{Sub}_\Delta}\sqrt{\frac{16c^2 H\phi_m^2\iota}{\Delta^2(s,x)}} + \sum_{(s,x)\notin\text{Sub}_\Delta}\frac{72c^2e^2\phi_m^2 H^4\iota}{\Delta(s,x)} + H \quad (96)$$

$$= \sum_{(s,x)\in\text{Sub}_\Delta}\frac{12c^2e^2\phi_m^2 H^3\iota}{\Delta(s,x)} + \sum_{(s,x)\notin\text{Sub}_\Delta}\frac{72c^2e^2\phi_m^2 H^4\iota}{\Delta(s,x)} + H \quad (97)$$

$$= \mathcal{O}\left(\sum_{(s,x)\in\text{Sub}_\Delta}\frac{\phi_m^2 H^3\log(SAT)}{\Delta(s,x)} + \sum_{(s,x)\notin\text{Sub}_\Delta}\frac{\phi_m^2 H^4\log(SAT)}{\Delta(s,x)}\right). \quad (98)$$

We can further simplify the regret bound by the fact that $\phi(\cdot) \leq H$ and arrive at the final bound,

$$\mathcal{O}\left(\sum_{(s,x)\in\text{Sub}_\Delta} \frac{\phi_m^2 H^3 \log{(SAT)}}{\Delta(s,x)} \sum_{(s,x)\notin\text{Sub}_\Delta} \frac{\phi_m^2 H^4 \log{(SAT)}}{\Delta(s,x)}\right) \tag{99}$$

$$= \mathcal{O}\left(\sum_{s,x\in\text{Sub}_\Delta} \frac{H^5 \log{(SAT)}}{\Delta(s,x)} + \sum_{s,x\notin\text{Sub}_\Delta} \frac{H^6 \log{(SAT)}}{\Delta(s,x)}\right). \tag{100}$$

$\square$

