# OpenReview forum: "Automatic Reward Shaping from Confounded Offline Data"
_ICML.cc/2025/Conference — ICML 2025 poster_

### Official Review · Reviewer_AQkY · 2025-03-12

**Overall Recommendation:** 3

**Summary:**

The paper aims to automate reward shaping when learning policies online via Reinforcement Learning (RL). Authors propose an automated approach for designing reward functions utilizing previously collected offline samples with unobserved confounders. State value upper bounds are calculated and used as a conservative optimistic estimation of the optimal state value function. These estimates are then plugged in to the Potential-Based formulation. Empirical results on toy tasks demonstrate that shaping function provides a better regret bound.

**Claims And Evidence:**

Please refer to strengths and weaknesses.

**Essential References Not Discussed:**

Please refer to strengths and weaknesses.

**Experimental Designs Or Analyses:**

Please refer to strengths and weaknesses.

**Methods And Evaluation Criteria:**

Please refer to strengths and weaknesses.

**Other Comments Or Suggestions:**

### Minors

* line 85: levering $\rightarrow$ leveraging
* line 97: denote the their $\rightarrow$ denote their
* line 210: optimal optimal $\rightarrow$ optimal
* line 210: state function $\rightarrow$ state value function
* line 216: form $\rightarrow$ from

**Other Strengths And Weaknesses:**

### Strengths
* The paper is theoretically rich and detailed in its explanations.
* The paper studies an important problem in the online setting, often overlooked by the RL community.

### Weaknesses
* **Empirical Evaluation:** While the paper makes a theoretical contribution, authors have provided empirical experiments to support the regret bound. My main concern is the empirical evaluation carried out on gridworld tasks which present a lower number of confounding states. State spaces for these tasks are countable and thus, it is non-trivial to study the effects of confounders in such a setting. Could the authors provide if they inject any noise in the data distribution? How much confounding does the agent encounter on average? It would be worthwhile to inspect the shaping function on varying grid sizes and under different settings of robustness, i.e- different start states or a larger challenging setting.
* **Comparison to Q-UCB:** I am struggling to understand the differences and contributions of the shaping scheme when compared to Q-UCB. Primarily, authors introduce zero initialization of Q-values, UCB bonus based on potential function and shaped reward during the updates. Could the authors explain their reasoning behind these choices? Also, could a comparison between the causal shaping function and Q-UCB be provided? In its current form, it is difficult to understand the improvements of utilizing PBRS.
* **Overall Presentation and Writing:** The presentation and writing quality of the paper could be further improved. Various portions of sections 2 and 3 could be made more intuitive. Theoretical explanations following theorems and corollaries could be made more concise and clear. Plots corresponding to experiments could be enlarged and reduced in filesize as the paper takes a long time to load. Grammatical errors could be reduced. A few are mentioned below.

**Questions For Authors:**

Please refer to strengths and weaknesses.

**Relation To Broader Scientific Literature:**

Please refer to strengths and weaknesses.

**Theoretical Claims:**

Please refer to strengths and weaknesses.

---

> ### Author Rebuttal · Authors · 2025-04-01
>
> We thank the reviewer for the detailed feedback and appreciate your recognition of the significance of our work. We have addressed each of your concerns in the responses below.
>
> > Weakness 1: “Could the authors provide if they inject any noise in the data distribution? How much confounding does the agent encounter on average? It would be worthwhile to inspect the shaping function on varying grid sizes and under different settings of robustness, i.e- different start states or a larger challenging setting.”
>
> This paper investigates reward shaping using confounded observational data and demonstrates how the learned shaping function can accelerate future online learning processes. Like many online reinforcement learning (RL) algorithms, the performance guarantee of our proposed method relies on the assumption that the measured variables are discrete (as in MDPs). To our knowledge, the evaluation environments we used are comparable to those in existing studies on online RL and reward shaping in terms of state space and complexity.
>
> Although the discrete state space may seem simple, reward shaping remains significantly challenging when unobserved confounding is present or cannot be assumed away. For instance, in our simulation shown in Figure 4(b), naively learning shaping functions from confounded observations, while ignoring the confounders, often fails to capture valuable information, resulting in inconsistent improvements in the performance of online RL algorithms. In contrast, our proposed causal approach achieves an order-of-magnitude improvement.
>
> In our current experimental setup, uniform random starting positions are used to sample offline data from the environment. We acknowledge that in a large state space with limited starting locations, the offline data may not sufficiently explore the state space, resulting in optimistic estimates for those under-explored states. However, this optimistic estimation is the most reasonable guess at the optimal values of those states, as no other information or assumptions can rule out the possibility that these under-explored states may be highly rewarding.
>
> > Weakness 2: “Could the authors explain their reasoning behind these choices? Also, could a comparison between the causal shaping function and Q-UCB be provided?”
>
> Thank you for the opportunity to clarify this issue! We have conducted experiments on vanilla Q-UCB, which corresponds to the “No Shaping” baseline in Fig. 4. In the revised version, we have updated the experiment legend to use the term “vanilla Q-UCB.”
>
> Regarding the design choices we made in Q-UCB Shaping:
> 1. **Zero Initialization of Q-values**: Zero initialization is used because when applying a shaping function that serves as an upper bound of the optimal state value, the optimal state values after shaping are shifted downward by the magnitude of the shaping function, which is now at most zero.
>
> 2. **Modification of the UCB Bonus**: We modify the UCB bonus to use potential functions because the standard UCB bonus applies an overly optimistic estimate (i.e., number of steps times the maximum reward per step). Since we now have a better upper bound, using it helps reduce overestimation.
>
> 3. **Shaped Reward During Updates**: Shaped rewards are used during updates to incorporate additional information extracted from offline data, thereby enhancing the online learning process.
>
> > Weakness 3: “Plots corresponding to experiments could be enlarged and reduced in file size as the paper takes a long time to load. Grammatical errors could be reduced.”
>
> We appreciate the reviewer’s careful reading of our work. We have corrected the typos, reduced image file sizes, adjusted figure sizes, and further polished the language in the revised version. We will upload it once the portal opens.

---

> > ### Comment · Reviewer_AQkY · 2025-04-02
> >
> > I thank the authors for their response to my comments. My concerns have been addressed and I would like to keep my current score. I thank the authors for their efforts.

---

> > > ### Author Response · Authors · 2025-04-03
> > >
> > > Thanks for the quick updates! We would like to answer further questions if there are other major blockers in raising the score. And as always, we highly appreciate your time and consideration!

---

### Official Review · Reviewer_89wj · 2025-03-13

**Overall Recommendation:** 3

**Summary:**

This paper addresses the challenge of designing reward shaping functions in reinforcement learning, particularly in the context of Potential-Based Reward Shaping, which traditionally relies on manually crafted potential functions. To overcome this limitation, the authors propose a data-driven approach that automatically learns reward shaping functions from confounded offline data using causal state value upper bounds. The study focuses on Confounded Markov Decision Processes, where unobserved confounders exist at each decision step, making standard offline reinforcement learning methods unreliable. The paper introduces a model-free UCB-based algorithm that leverages derived shaping functions to improve learning efficiency and provides the first theoretical gap-dependent regret bound for PBRS in model-free settings. Finally, empirical results support the theoretical findings.

**Claims And Evidence:**

The paper claims that Q-UCB Shaping improves sample efficiency, enables convergence to the optimal interventional policy, and provides theoretical guarantees for its causal upper bounds. These claims are partially well-supported.
The causal Bellman bounds provide a solid theoretical foundation, making the claim about upper bounding optimal state values convincing. The empirical results show better performance compared to baselines in confounded environments, supporting claims of improved sample efficiency.
However, convergence to the optimal interventional policy is assumed rather than directly analyzed. Additional policy divergence or optimality gap analyses would strengthen this claim.

**Essential References Not Discussed:**

N/A

**Experimental Designs Or Analyses:**

The experiment settings clearly simulate confounding effects in offline reinforcement learning, and the comparison against multiple reward shaping strategies is insightful.
However, the environments are relatively small-scale, limiting their real-world applicability.
Additionally, the study would be more compelling if it included experiments with more complex confounding structures, such as hierarchical or multi-agent confounders.

**Methods And Evaluation Criteria:**

The use of CMDPs and causal bounds is well-motivated for the problem of learning from confounded offline datasets. The choice of MiniGrid windy environments is reasonable for demonstrating confounding effects.
However, the limited scale of the environments makes it unclear how the method generalizes to high-dimensional settings.
The paper does not compare against stronger offline RL baselines, such as BCQ, CQL, or IRM-based methods, which would provide a better context for the performance gains.

**Other Comments Or Suggestions:**

I acknowledge that there are aspects of this paper that I am not fully familiar with, and I may not be a fully qualified reviewer for this topic. However, I am very willing to discuss the work with the authors.

**Other Strengths And Weaknesses:**

**Strengths**
1. The writing is clear and well-structured.
2. The introduction of causal Bellman optimality bounds is a valuable contribution.
3. Combining UCB-based exploration with causal bounds is an interesting and underexplored direction.

**Weaknesses**
1. The approach is tested in small grid-worlds but not in larger RL settings.
2. The paper does not compare against state-of-the-art offline RL methods, such as CQL, BCQ or IQL.

**Questions For Authors:**

1. Can you provide empirical evidence showing how close the learned policy is to the optimal interventional policy? Policy divergence metrics would help clarify this.
2. How well does the method scale to continuous or large state-action spaces? Have you considered testing in Mujoco or other RL benchmarks?
3. How does the performance change when the offline dataset is sparse or comes from suboptimal behavior policies?

**Relation To Broader Scientific Literature:**

The paper connects to:
1. Offline RL: The method is related to batch-constrained RL and robust RL, but does not compare against conservative Q-learning or implicit Q-learning.
2. Exploration via UCB: The paper applies UCB exploration in CMDPs, but prior works (e.g., bandit-based causal RL) should be discussed to highlight novelty.

**Theoretical Claims:**

The paper presents three key theoretical results: (1) the Causal Bellman Optimality Equation (Theorem 3.1), which derives an upper bound for the optimal value function in CMDPs, (2) the Unified Causal Optimal Upper Bound (Corollary 3.2), which extends the bound when multiple offline datasets are available, and (3) the Regret Bound for Q-UCB Shaping (Theorem 4.5), showing improved regret bounds. While these proofs appear correct, they have some limitations.
1. They assume a tight upper bound on state values, which may not always hold in practice.
2. They do not explicitly analyze the impact of latent variable shifts under partial observability.

---

> ### Author Rebuttal · Authors · 2025-04-01
>
> We are grateful for your detailed comments. We have addressed your concerns in the sequel.
>
> > Optimality of the learned policy & Question 1
>
> While we adopt the regret analysis framework, we acknowledge that it offers only a weaker guarantee of convergence to the optimal policy. A PAC framework may be more suitable for optimality analysis. For empirical evidence of the learned policy quality, we have visualizations in the experiment section (Fig. 5). Since the learned policy is deterministic, we further report the ratio of states where the learned policy is optimal, averaged over three seeds with values closer to 1 indicating better performance.
>
> algo/env|Windy Empty World|LavaCross(easy)|LavaCross(hard)|LavaCross(maze)
> -|-|-|-|-
> Vanilla Q-UCB| .0 | .0 | .0 | .03
> Q-UCB Shaping (Ours)|.76|**.70**|**.81**|**.90**
> Shaping w/ Min. Val.|.72 |.49 | .30 | .24
> Shaping w/ Max Val.|.73 |.52 | .59 | .33
> Shaping w/ Avg. Val.|.74 |.50 | .49 | .24
> BCQ| .79|.50|.67|.50
> Shaping + BCQ| **.88**|.31|.22|.03
>
> > Applicability to high-dimensional and continuous envs & Weakness 1 & Question 2
>
> Thanks for the question! You can refer to the first part of our response to reviewer dtkk.
> > Tight upper bounds
>
> Our proposed method is data-driven, so the learned bound in practice may not be tight. Bound tightness can be improved when multiple datasets are available by taking the minimum over those. Theoretically, the tightness is determined by the natural bounds, without requiring further assumptions about the environment. To illustrate this, we provide a simpler example involving bandits to demonstrate how the bound on rewards can be tight. Our proposed bounds in Equations 5 and 6 are conditional extensions of this idea.
>
> Example: Consider a confounded MAB where reward $Y$ depends on action $X$ and they are confounded by a variable $U$. All variables are binary. The natural bound indicates that the distribution of the reward under interventions can be bounded as $P(y|do(x))=P(x,y)+\sum_u P(y|x,u)(P(u)-P(u,x))\leq P(x,y)+\sum_uP(u)-P(u,x) = P(x,y)+1-P(x)$. Equality is achieved and the bound is tight when $u=0,1,P(y|x,u)=1$, which is possible if the reward is deterministic given an action.
>
> > Latent variable distribution shifts
>
> In this paper, we study the problem of reward shaping in confounded MDPs. To the best of our knowledge, this work is novel and extends reward shaping methods to address the practical challenges of confounders. We acknowledge that reward shaping in confounded POMDPs and under distribution shifts is a challenging and exciting research question. We appreciate the suggestion.
>
> > More complex confounding biases structures
>
> Since our work is built upon the framework of CMDPs, we do not impose additional constraints on how confounders may correlate with each other. As a result, our approach can handle hierarchical confounders when they fit within the causal diagrams of CMDPs. Exploring the multi-agent case is an interesting direction for future work, as this paper focuses on designing shaping functions for training single agents.
>
> > Related works:
>
> We have extended the discussion on both offline RL and causal bandits in the revised version. Briefly, many existing offline RL methods learn q-values without accounting for confounders. Some approaches involve learning initial values offline and then fine-tuning them online, but this offline initialization often diminishes during online learning. Our proposed method addresses confounders in the offline data and preserves the extracted knowledge during the online phase through reward shaping.
>
> Regarding causal bandits, our approach shares the same motivation of using causal bounds to address confounding biases and reduce the regret. However, we extend this idea by combining bounds with UCB exploration in sequential domains and applying it to the optimal Bellman equation.
>
> > Weakness 2: Compare with SOTA offline baselines
>
> We have added experiments to the revised version comparing our method with BCQ, and we summarize the state ratio with optimal policies in the table provided in the first part of our response. We train BCQ using the same mixed confounded datasets as our method’s inputs. We evaluate the performance of the policy extracted directly from BCQ q-values, and the performance of Q-UCB using BCQ values as shaping functions. Our method outperforms both BCQ baselines in non-empty windy grid worlds, where confounder information is essential for completing the tasks.
>
> > Question 3: performance change under imperfect offline data
>
> In the revised version, we provide performance results when the good behavioral policy is excluded. To summarize, the final regrets remain nearly unchanged [[img link]](https://ibb.co/album/JBYSZm). Because the bounds from optimal demonstrations are typically overly optimistic; When unifying such bounds with others from imperfect demonstrations, they are not selected by the min operator and thus have minor effects on the shaping functions.

---

### Official Review · Reviewer_dtkk · 2025-03-14

**Overall Recommendation:** 3

**Summary:**

This paper focuses on the automated design of a reward shaping function from offline data, potentially contaminated by unobserved confounding bias. The authors propose leveraging causal state value upper bounds from offline datasets as a conservative, optimistic estimate of the optimal state value, which is then used as a state potential in Potential-Based Reward Shaping (PBRS) to achieve a more favorable gap-dependent regret bound. Experimental results demonstrate the effectiveness of the proposed approach.

**Claims And Evidence:**

Yes.

**Essential References Not Discussed:**

N/A

**Experimental Designs Or Analyses:**

The expeimental environment support the fucused challenge well. The proposed method outperform baselines and the provided visualization of the policy make sense to me.

The only one concern is how would the proposed method in more realistic environments, like MuJoCo?

**Methods And Evaluation Criteria:**

Yes. The proposed method demonstrates strong potential for supporting offline reinforcement learning with reward shaping, and the evaluation benchmark aligns well with the primary focused challenges.

**Other Comments Or Suggestions:**

In Line 210, it seems a copy-paste error here.

**Other Strengths And Weaknesses:**

**Strength**

- The paper is well written and easy to follow.
- The problem is interesting and important to me.
- The examples provided are very helpful for my understanding.

**Weakness**

However, I am somewhat concerned about the generality of the setting. As it is hard to say in the realistic task, how the confounders exist, like between state and action, or only action and reward. Could you please provide additional real-world examples to illustrate its broader applicability?

Furthermore, it would be beneficial if Figure 2 included the transition functions and a clearer demonstration of how the unobservable confounder operates, as this would enable a quicker and more thorough comprehension.

Lastly, I would like to confirm my understanding: the confounders can be viewed as hidden states of the environment. Consequently, not only must the behavior policy account for these confounders, but the learned policy should also incorporate them when making decisions. Is that right? Could you please also clarify the role of the confounders in decision-making and policy learning? Is it possible to learn the optimal policy without observing the confounders?

**Questions For Authors:**

- Could you provide more real-world examples or case studies to illustrate the broader applicability of your proposed method, beyond the standard benchmark environments?

- In which scenarios are confounders critical for policy learning, and in which cases might they be less relevant or unnecessary? How should one identify and account for such confounders in practice? Can I regard it as one kind of Partitial Observation?

**Relation To Broader Scientific Literature:**

N/A

**Theoretical Claims:**

Yes.

---

> ### Author Rebuttal · Authors · 2025-04-01
>
> We thank you for your thoughtful feedback. We appreciate your acknowledgment of the significance of our work and have addressed your concerns in the sequel.
> > “The only one concern is how would the proposed method in more realistic environments, like MuJoCo?”
>
> In this paper, we primarily focus on learning shaping functions from observational data and using the learned functions to accelerate online learning processes. Like many online RL algorithms, we use UCB as our basic learning method. The theoretical guarantees of UCB’s performance require the state-action space to be discrete, which may not directly apply to continuous environments like MuJoCo. However, this is a common challenge for most online RL algorithms. Extending these to continuous domains is an exciting challenge, and we appreciate the suggestion.
> That said, we would like to briefly explain how our proposed method can be extended to high-dimensional and continuous state/action domains. For each component in our causal bound – i.e., policy distribution, reward, transitions, and values – we can train neural networks to learn these from offline data. The primary challenge is that the max operator of the bound in high-dimensional and continuous spaces cannot be computed exactly, necessitating approximation. Specifically, we can train a separate neural network to select the next possible best state based on observational data and then extract values from the learned value networks. This is a natural extension of our contributions; however, we were unable to provide more formal results at this point. Thus, we leave these out of the current manuscript.
>
> > “Could you provide more real-world examples or case studies to illustrate the broader applicability of your proposed method?”
>
> Unobserved confounders naturally arise when decision-makers consider factors that a passive observer cannot access. For instance, when training autonomous vehicles, data is often collected from engineering prototypes equipped with costly laser radars. However, to reduce production costs, we may wish to train less expensive vehicles equipped only with cameras using this data. Here, radar information becomes an unobserved confounder for the learner. Similarly, in robotics, numerous factors influence action outcomes without being monitored, contributing to what is known as the “sim-to-real gap”, caused by unobserved confounding. For example, friction and instability in demonstrators’ joints may be unmonitored, affecting their behaviors. The method proposed in this work can be used to design denser reward signals when training robots or autonomous vehicles, using confounded data as such.
>
> > “Furthermore, it would be beneficial if Figure 2 included the transition functions and a clearer demonstration of how the unobservable confounder operates, as this would enable a quicker and more thorough comprehension.”
>
> In the robot walking example, the step size ($U_t$) required to stabilize its body (($F_t=0 \rightarrow F_{t+1}=1}$)) is an unobserved confounder. We will update the manuscript to clarify this issue. Thank you for the suggestion.
>
> > “In which scenarios are confounders critical for policy learning, and in which cases might they be less relevant or unnecessary? How should one identify and account for such confounders in practice? Can I regard it as one kind of partial observation?”
>
> Confounders can be taken as partial observations, except that behavioral policies may access some of them. Consequently, the learned policy should account for these – though, as we will explain next, there are cases where this may not be necessary. However, the unobserved confounders are not directly accessible to the learned policy and require special treatment, which is the objective of the causal bounds proposed in this work.
> Regarding optimality, as demonstrated in our experiments, the learned policy proves to be a safer choice compared to one that aims to collect all coins and reach the goal location. This latter policy is the best possible outcome the agent could achieve without observing the wind. However, it is no longer optimal when compared to a behavioral agent with greater sensory capabilities. Generally, when unobserved confounders exist, it is understood in the literature (Pearl, 2009) that the optimal policy is under-determined by observational data (not identifiable) without additional experiments or assumptions.
> Interestingly, the current models usually assume away the confounders, and not the other way around. In other words, our goal is not to learn the unobserved confounders, since they are unobserved, but to have methods that are robust and could protect against them (without having to measure them explicitly). We argue in this paper that it is hard to assume away confounders in many real-world domains and propose solutions to it. In a way, we are relaxing something baked in many methods in the literature.
> > Typos
>
> Thanks! We will fix it in the revised version.

---

### Decision · Program_Chairs · 2025-05-01

**Decision:**

Accept (poster)

**Comment:**

This paper tackles the important problem of reward shaping in the presence of unobserved confounding. The authors propose an approach that constructs conservative, optimistic reward shaping functions by leveraging causal state value upper bounds derived from offline datasets. This method is then applied within the PBRS framework to yield improved gap-dependent regret bounds. All reviewers appreciated the clarity of the paper and its strong theoretical foundation, particularly the derivation of causal Bellman optimality bounds and the integration of UCB exploration with causal reasoning. On the other hand, the reviewers consistently expressed concerns about the limited empirical evaluation and the comparison of the existing benchmarks, which would be better highlight in the future, despite the theoretical background of this paper.